# Chaperone dependency during biogenesis does not correlate with chaperone dependency during refolding

Divya Yadav [1], İdil I Demiralp [1,3], Mark Fakler [1] & Stephen D Fried [1,2 ✉]

## Abstract

**Many proteins require molecular chaperones to fold into their functional native forms. However, the roles of chaperones during primary biogenesis in vivo can differ from the functions they play during in vitro refolding experiments. Here, we use limited proteolysis mass spectrometry (LiP-MS) to probe structural changes incurred by the *E. coli* proteome when two key chaperones, trigger factor and DnaKJ, are deleted. While knocking out DnaKJ induces pervasive structural perturbations across the soluble *E. coli* proteome, trigger factor deletion only impacts a small number of proteins' structures. Overall, proteins which cannot spontaneously refold (or require chaperones to refold in vitro) are *not* more likely to be dependent on chaperones to fold in vivo. We find that chaperone-nonrefolders (proteins that cannot refold even with chaperone assistance) do not generally require chaperones to fold in vivo, strengthening the view that chaperone-nonrefolders are obligate co-translational folders. Hence, for some *E. coli* proteins, the vectorial nature of co-translational folding is the most important "chaperone".**

**Keywords** Chaperones; DnaK; Trigger Factor; Co-translational Folding; Structural Proteomics
**Subject Categories** Proteomics; Structural Biology

## Introduction

Many proteins must fold into specific three-dimensional structures to perform their native functions (Karplus, 2011; Dill and MacCallum, 2012; Dill et al, 2008). While the primary amino acid sequence generally encodes the information necessary to locate these native states, some proteins cannot fold spontaneously (Jumper et al, 2021; Anfinsen, 1973; Dobson et al, 1998; Braselmann et al, 2013; Baker, 2000). Particularly within the crowded intracellular environment, molecular chaperones often play critical roles in guiding nascent polypeptides toward

productive folding pathways, shielding aggregation-prone intermediates, and reverting kinetically trapped misfolded states that may be populated (Bukau et al, 2000; Mayer and Gierasch, 2019; Kim et al, 2013; Santra et al, 2017; Balchin et al, 2016; Wales et al, 2024; Clark, 2004; Bukau and Horwich, 1998; Jahn and Radford, 2005; Kampinga and Craig, 2010; Houry, 2001).

In *Escherichia coli*, three primary cytosolic chaperone systems cooperate to assist protein folding: Trigger Factor (Tig) (Hoffmann et al, 2010; Kaiser et al, 2006b; Oh et al, 2011; Huang et al, 2000; Martinez-Hackert and Hendrickson, 2009; Maier et al, 2005), DnaK/DnaJ (Hsp70/Hsp40) (Calloni et al, 2012; Schröder et al, 1993; Mayer and Bukau, 2005; Laufen et al, 1999; Mayer and Gierasch, 2019; Imamoglu et al, 2020; Rosenzweig et al, 2019; Bukau and Horwich, 1998), and GroEL/ES (Singh et al, 2020; Goloubinoff et al, 1989a; Farr, 2003; Brinker et al, 2001). Trigger Factor is an abundant ATP-independent holdase with peptidyl-prolyl isomerase activity that binds nascent peptide chains emerging from the ribosomal exit tunnel, first identified as being important for outer membrane protein biogenesis (Hoffmann et al, 2012; Merz et al, 2008; Crooke and Wickner, 1987; Kaiser et al, 2006a). The DnaK-DnaJ system, in collaboration with the nucleotide exchange factor GrpE, functions both co- and post-translationally, and has many activities: It facilitates primary biogenesis particularly of multi-domain proteins, facilitates hand-over to other chaperone systems (like the Hsp100 ClpB), and is upregulated during heat shock and other stressors (Mogk et al, 2015; Mayer, 2021; Baler et al, 1992; Wallace et al, 2015; Nussenzweig et al, 1997; Nollen and Morimoto, 2002; Craig et al, 1993; Clerico et al, 2019). GroEL/ES is the canonical type I chaperonin complex; it provides an isolated chamber that promotes folding of unfolded or misfolded proteins and is also essential for viability in *E. coli* (Fujiwara et al, 2010; Singh et al, 2020; Apetri and Horwich, 2008; Thirumalai et al, 2020; Brinker et al, 2001; Ying et al, 2005; Farr, 2003; Goloubinoff et al, 1989b).

While these chaperone systems have distinct mechanisms, they exhibit considerable functional overlap (Deuerling et al, 2003; Bhandari and Houry, 2015). For instance, Tig and DnaKJ can both be deleted individually, though not simultaneously under normal growth conditions (Teter et al, 1999), showing that their activities can compensate for one another (Genevaux et al, 2004; Wruck et al, 2018; Deuerling et al, 2003). This dynamic, cooperative network

[1]Department of Chemistry, Johns Hopkins University, Baltimore, MD 21218, USA. [2]T. C. Jenkins Department of Biophysics, Johns Hopkins University, Baltimore, MD 21218, USA. [3]Present address: Skaggs Institute for Chemical Biology, Scripps Research, La Jolla, CA 92037, USA. ✉E-mail: sdfried@jhu.edu

enables *E. coli* to maintain proteome stability and adapt to diverse environments.

Despite detailed mechanistic studies on individual chaperones, comprehensive proteome-wide analyses of folding dependencies in vivo remain relatively limited. Proteome-wide methods have been used to explore chaperone activities (Kerner et al, 2005; Calloni et al, 2012; Fujiwara et al, 2010; Niwa et al, 2012; Fujiwara et al, 2017) primarily by identifying proteins that co-precipitate with chaperones upon stabilizing the complexes. Whilst powerful, these approaches do not necessarily show which clients *depend* on the chaperone to fold, since it is possible for a chaperone to interact with substrates that do not require it (To et al, 2022; Kim et al, 2013; Bukau and Horwich, 1998). Our group has applied a structural proteomic approach to assess which proteins in *E. coli* can spontaneously return to their native structures after being first denatured and then diluted from denaturant in vitro (To et al, 2021). These experiments showed that certain factors, like fold-types, protein isoelectric point (pI), and number of domains correlate well with reversible refoldability. Moreover, these experiments were expanded to include *E. coli* chaperones GroEL/ES and DnaKJ to establish which proteins require chaperone assistance to refold from their fully denatured forms (To et al, 2022). However, these experiments do not provide information about the role of *E. coli* chaperones during primary protein biogenesis, which occurs co-translationally. There is increasing appreciation that outside of stress conditions, most chaperone activity may be occurring in a co-translational manner (Shieh et al, 2015; Willmund et al, 2013; Oh et al, 2011; Ciryam et al, 2013; Roeselová et al, 2024a; Wales et al, 2024; Roeselová et al, 2024b; Fedorov and Baldwin, 1997; Houry, 2001). In addition, cells very rarely see fully denatured protein conformations of the kind that would be populated by high concentrations of denaturant, even during acute heat shock, because sub-lethal heat shock temperatures are still considerably lower than typical $T_m$'s of many mesophilic proteins (Schopper et al, 2017; Leuenberger et al, 2017; Jarzab et al, 2020).

To address this gap, we applied the structural proteomic method limited-proteolysis mass spectrometry (LiP-MS) to systematically determine structural perturbations that accrue across the *E. coli* proteome in strains lacking non-essential chaperones (Δ*dnaKJ* and Δ*tig*). These experiments effectively allow us to determine which proteins endure a permanent structural change if they are biosynthesized in a background that lacks specific protein folding machinery, and to explore their relationship to growth temperature.

We find that DnaKJ is important for many *E. coli* proteins to navigate to their native conformations at 30 °C, but curiously, less so at 37 °C. Moreover, the temperature dependence is particularly acute for proteins which bind to cofactors. We propose a biophysical model for the "cold-dependency" of a certain subset of proteins on DnaK. Trigger factor is another important chaperone in *E. coli*'s proteostasis network, but we find that its deletion leaves *very* few proteins structurally perturbed. This finding suggests that whilst trigger factor may increase the yield or efficiency of protein folding, proteins that engage trigger factor can ultimately locate their native conformations without it, likely by relying on other chaperones.

As LiP-MS is an emerging structural proteomic method (Feng et al, 2014; Schopper et al, 2017; de Souza and Picotti, 2020) and there are multiple ways to analyze its data (Manriquez-Sandoval et al, 2024; Nagel et al, 2025), it is important to critically assess whether the structural differences LiP-MS measurements report on can be used to make functional predictions. We show that PGK, when isolated from different chaperone knock-out strains, is biochemically and biophysically identical to the isolate from wild-type *E. coli*, consistent with the absence of a significant LiP signal for this protein. In addition, we observe deficits in activity levels for another enzyme (catalase) that we predicted to be structurally altered in a functionally relevant manner in Δ*dnaKJ* cells according to LiP-MS.

Lastly, we carefully compare the experiments reported here on the dependence of certain proteins on DnaKJ and Tig when they are folded in vivo to a reanalyzed dataset interrogating chaperone dependency for proteins to refold in vitro from a chemically denatured state. We find that these subsets of proteins show little overlap. Moreover, features associated with proteins which cannot spontaneously refold are quite distinct from the features associated with proteins that we find are DnaKJ-dependent or Tig-dependent. Hence, our study demonstrates that the role chaperones play during primary biogenesis and during in vitro refolding are likely quite different and moreover point to co-translational folding as being a significant chaperone in its own right.

## Results

### The effect of deleting DnaKJ and tig on the structure of the *E. coli* proteome

To study how chaperone deletion affects protein structure in *E. coli*, we performed limited proteolysis (LiP) followed by mass spectrometry (LiP-MS) on extracts derived from Δ*dnaKJ* and Δ*tig* knock-out strains and compared them to the parental strain, MC4100 (Fig. 1A). Proteins from wild-type (WT), Δ*dnaKJ*, and Δ*tig* strains were pulse proteolyzed with proteinase K (PK) to probe solvent-accessible regions, then digested with trypsin, and the resulting peptide fragments were quantified by LC-MS/MS. Structural changes were inferred by comparing the abundance of half-tryptic peptides between the chaperone knockouts and WT samples. Mass spectra were searched and quantified using FragPipe (Yu et al, 2021; Kong et al, 2017), and downstream analysis for LiP-MS (including FDR correction) was implemented in FLiPPR (Manriquez-Sandoval et al, 2024). To set a conservative cut-off for which proteins were deemed structurally perturbed relative to WT, we required that a protein possess two or more cut-sites with significantly different PK susceptibility in the knock-out strain (> twofold change in abundance after normalizing for protein abundance, *P* value by Welch's *t* test <0.05 following protein-wise Benjamini–Hochberg FDR correction). Analysis of the individual replicates showed satisfactory data quality (Fig. EV1), with consistent numbers of proteins, peptides, and peptide-spectrum matches between strains and replicates across the board (Fig. EV1A–C,E–G), low levels (~10%) of half-tryptic peptides in trypsin-only control samples, and higher levels (~55%) of half-tryptic peptides in PK-treated samples (Fig. EV1D,H). Peptide volcano plots displayed tripartite structure (Fig. EV1I–L) with most peptides (ca. 20,000 per experiment) occupying a central region with no significant change and two smaller lobes (with 1000–2000 peptides each) corresponding to sites where PK could cut only in

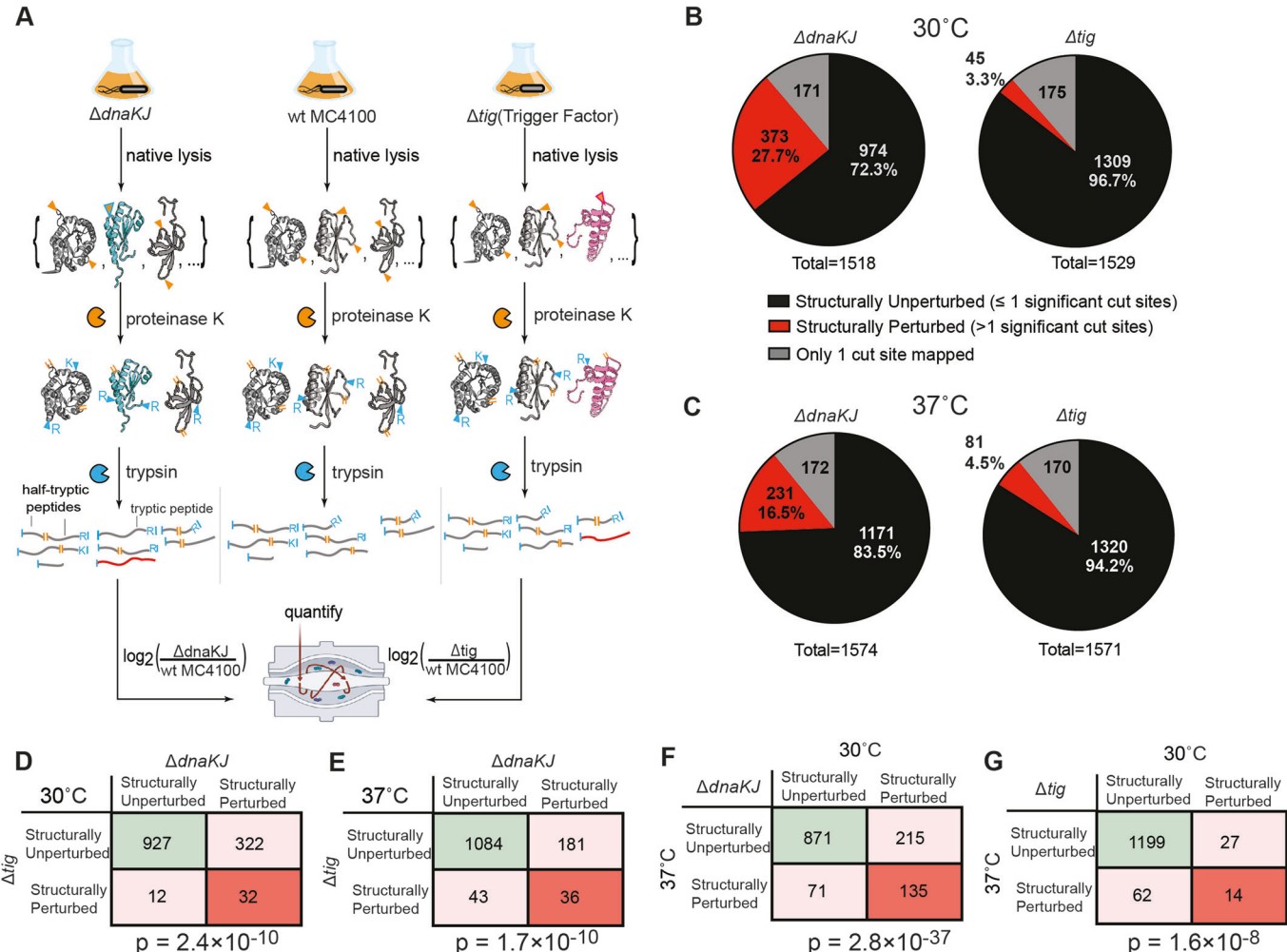

**Figure 1. LiP-MS to assess the role of trigger factor, DnaKJ on the structural integrity of the *E. coli* proteome.**

(A) Schematic overview of the LiP-MS workflow applied to investigate proteome-wide structural alterations in *E. coli* MC4100 wild-type (WT), Δ*dnaKJ*, and Δ*tig* strains. Lysates from each strain were prepared under native conditions, and protein structures were interrogated via pulse proteolysis using proteinase K. Structural changes with respect to WT were probed by label-free quantification of the resulting peptide fragments (three biological replicates). (B, C) Pie charts displaying the proportion of structurally perturbed (red) and structurally unperturbed (black) proteins in Δ*dnaKJ* and Δ*tig* strains at 30 °C (B) and 37 °C (C). Proteins with only one peptide mapped to them are shown in gray and were excluded from further analyses. The total numbers of identified proteins are indicated for each condition (see Source Data for Fig. 1B,C for summary data for all proteins assessed under all conditions). (D–G) Contingency tables showing the number of proteins that are structurally perturbed under both conditions, one condition, or not structurally perturbed under either condition when cross-correlating (D) the Δ*dnaKJ* and Δ*tig* datasets both at 30 °C; (E) the Δ*dnaKJ* and Δ*tig* datasets both at 37 °C; (F) the effect of Δ*dnaKJ* at 30 °C vs. 37 °C; (G) the effect of Δ*tig* at 30 °C vs. 37 °C. P values are calculated using Fisher's exact test (see Source Data Fig. 1D–G). Source data are available online for this figure.

either the WT or knock-out strain. In general, the lobes possessed a similar number of tryptic and half-tryptic peptides, suggesting that chaperone deletion does not necessarily result in more disordered or less-packed conformations.

At 30 °C, Δ*dnaKJ* cells displayed widespread disruption: 27.7% (373 out of 1347) of proteins showed perturbations to their PK susceptibility conformations (Fig. 1B, left; Source Data Fig. 1B,C). This is broadly consistent with many studies highlighting the multifaceted function of Hsp70 in *E. coli* (Rosenzweig et al, 2019; Clerico et al, 2019). In contrast, only 45 proteins (3.3%) were structurally perturbed in the Δ*tig* strain under the same conditions

(Fig. 1B, right; Source Data Fig. 1B,C). Hence, even though trigger factor is an important chaperone that engages many *E. coli* nascent chains (Oh et al, 2011; Haldar et al, 2017; Martinez-Hackert and Hendrickson, 2009), these LiP-MS data suggest that its absence can be more easily compensated for by other members of the proteostasis network. Many of these proteins were *also* structurally perturbed in Δ*dnaKJ* as well (Fig. 1D), suggesting that they are demanding proteins to properly fold in general. The contingency table in Fig. 1D (and to a lesser extent in Fig. 1E) highlights how much more important DnaKJ is than Tig for achieving native-like protein structures: Whereas 322 proteins are perturbed when

DnaKJ is deleted but structurally unaffected when Tig is deleted, the inverse is only true for 12 proteins. This result contrasts somewhat with in vitro experiments that have shown Tig to be a potent chaperone, for instance, in promoting the refolding of GAPDH (Oh et al, 2011; Wu et al, 2022; Huang et al, 2000); in vivo however its function can evidently be replaced by other members of the E. coli's proteostasis network.

We find that deleting DnaKJ at 30 °C results in a concomitant upregulation of other chaperones (GroEL, IbpB, IbpA, SurA) and proteases (Lon, ClpP, HslV), with many other (277) E. coli proteins' abundances going significantly down (>twofold, Benjamini–Hochberg adjusted P value < 0.05; Fig. EV2A). On the other hand, fewer chaperones and proteases have their levels dramatically altered when trigger factor is knocked out, and only 59 proteins' abundances significantly decrease (Fig. EV2A,B). Because chaperone deletion causes the proteostasis network to rebalance (e.g., >fourfold higher levels of GroEL in ΔdnaKJ), we note that the structural perturbations we observe here are those that persist even after other chaperones and proteases are upregulated, implying their native conformations require DnaKJ/trigger factor in a manner that is not readily complemented by other proteostasis factors. The structural effect of deleting DnaKJ does not show many interesting overlaps with gene ontologies (probably because it plays quite a universal role), though Tig deletion results in targeted changes to proteins involved in insertion at the membrane and outer membrane assembly, consistent with prior literature (Fig. EV2C,D) (Crooke and Wickner, 1987).

Curiously, we found that at 37 °C, the disruption to the proteome imparted by ΔdnaKJ is smaller, with now only 16.5% (231 out of 1402) proteins showing structural alterations (Fig. 1C, left). Cross-correlating the two datasets and counting only proteins that were confidently assessed in both conditions (Fig. 1F), we indeed find 215 proteins that are structurally perturbed in ΔdnaKJ only at 30 °C but not at 37 °C. This is surprising because DnaK is a heat-shock protein, typically thought to be more functionally important at elevated temperature (Lindquist and Craig, 1988; Parsell et al, 1994; Glover and Lindquist, 1998; Baler et al, 1992; Wallace et al, 2015; Nollen and Morimoto, 2002; Parsell et al, 1993; Craig et al, 1993). Trigger factor, on the other hand, displays the more expected behavior of becoming more important for native protein structures formation at 37 °C (cf. Fig. 1C, right, G), with 81 proteins becoming structurally perturbed in its absence.

We also examined whether temperature alone alters protein structure in the wild-type background. Comparing WT proteomes at 30 °C and 37 °C by LiP-MS revealed 66 proteins out of 1434 (4.6%) with altered PK susceptibility, each possessing two or more significant cut-sites (Appendix Fig. S1A,B; Source Data Fig. S1). The most highly perturbed was RaiA, a ribosome hibernation factor, with 19 altered cut-sites, consistent with its role in modulating ribosome function under environmental changes. Other affected categories included RNA maintenance factors (DeaD [also called cold-shock DEAD-box protein] and Rne), three chaperones (DnaK, ClpB, HtpG), and seven proteins involved in polyatomic anion redox chemistry.

Taken together, these findings demonstrate that DnaKJ plays a critical role in shaping the structures of the native E. coli proteome under unstressed conditions, whereas trigger factor ostensibly plays a more specialized role that is also more easily compensated for by other chaperones.

## Structural features that correlate with chaperone dependence

Focusing on the 30 °C dataset, we grouped proteins into categories based on various protein features, such as the number of domains it hosts, its isoelectric point (pI), and which folds and cofactors it contains (Fig. 2), and assessed their relationships with structural perturbation in the chaperone knock-out strains (Source Data Fig. 2C–I). Naively, one might think that proteins that are more dependent on chaperones to assume their native structures may also be less capable of spontaneously refolding, so we compared these results to earlier data (To et al, 2021) (reprocessed using the same data processing workflow used here (Manriquez-Sandoval et al, 2024; Data ref: Manriquez-Sandoval et al, 2024)) on proteome-wide refoldability. In contrast to this expectation, we found that the loss of DnaKJ affects quite a distinct set of proteins compared to the nonrefolders (Fig. 2A; Source Data Fig. 2A,B). More specifically, 35% of DnaK-dependent proteins (31% at 37 °C) are nonrefolding, which is even less than the overall frequency of nonrefoldability. Hence, DnaK-dependent proteins are not more likely to be nonrefoldable in vitro; similarly, neither were Tig-dependent proteins (Fig. 2B; Source Data Fig. 2A,B). In the following analysis, we will focus on comparing refoldability to structural perturbations in ΔdnaKJ, as proteins which were structurally perturbed in Δtig did not display strong correlations with the protein features considered (Fig. 2, blue traces), most likely because it is a rather small set.

One trait that showed qualitative consistency was the number of globular domains (Manriquez-Sandoval and Fried, 2022). Overall, increasing the number of domains in a protein increases its dependency on DnaKJ and its likelihood of not spontaneously refolding (Fig. 2C) consistent with the accepted understanding that non-native interdomain contacts can be a common cause of misfolding, are more common with more domains, and can be prevented by DnaK (Zheng et al, 2013; Imamoglu et al, 2020).

However, the similarity between in vitro refoldability and DnaKJ-dependence is limited. For instance, refoldability is lowest for mildly acidic proteins (Fig. 2D). On the other hand, loss of DnaKJ does not have a major relationship with pI (Fig. 2D, red trace). Perhaps the most striking difference is seen when we consider the fold types of the constituent domains, which we identify using the ECOD system (Cheng et al, 2014). Previously, we noted that there is a large dispersion in the inherent refoldability of different domains, with some topologically complex domains not refolding efficiently (Fig. 2E, gray trace). On the other hand, fold type only weakly explains DnaKJ-dependence (Fig. 2E, red trace). The only fold that seems to have notably high dependence on DnaKJ is the thiamin diphosphate (THDP)-binding domain, but other challenging-to-refold domains were not structurally perturbed in ΔdnaKJ.

Using data on EcoCyc (Karp et al, 2025, 2018; Keseler, 2004), we also considered cofactors associated with each protein and found DnaKJ dependence was once again quite distinct from nonrefoldability (Fig. 2F). Previously, we noted that iron–sulfur cluster proteins were amongst the most refoldable, and the least refoldable proteins tended to involve non-covalently bound metal cofactors. In contrast, the proteins most dependent on DnaKJ were those involving covalently bound factors, such as iron–sulfur clusters,

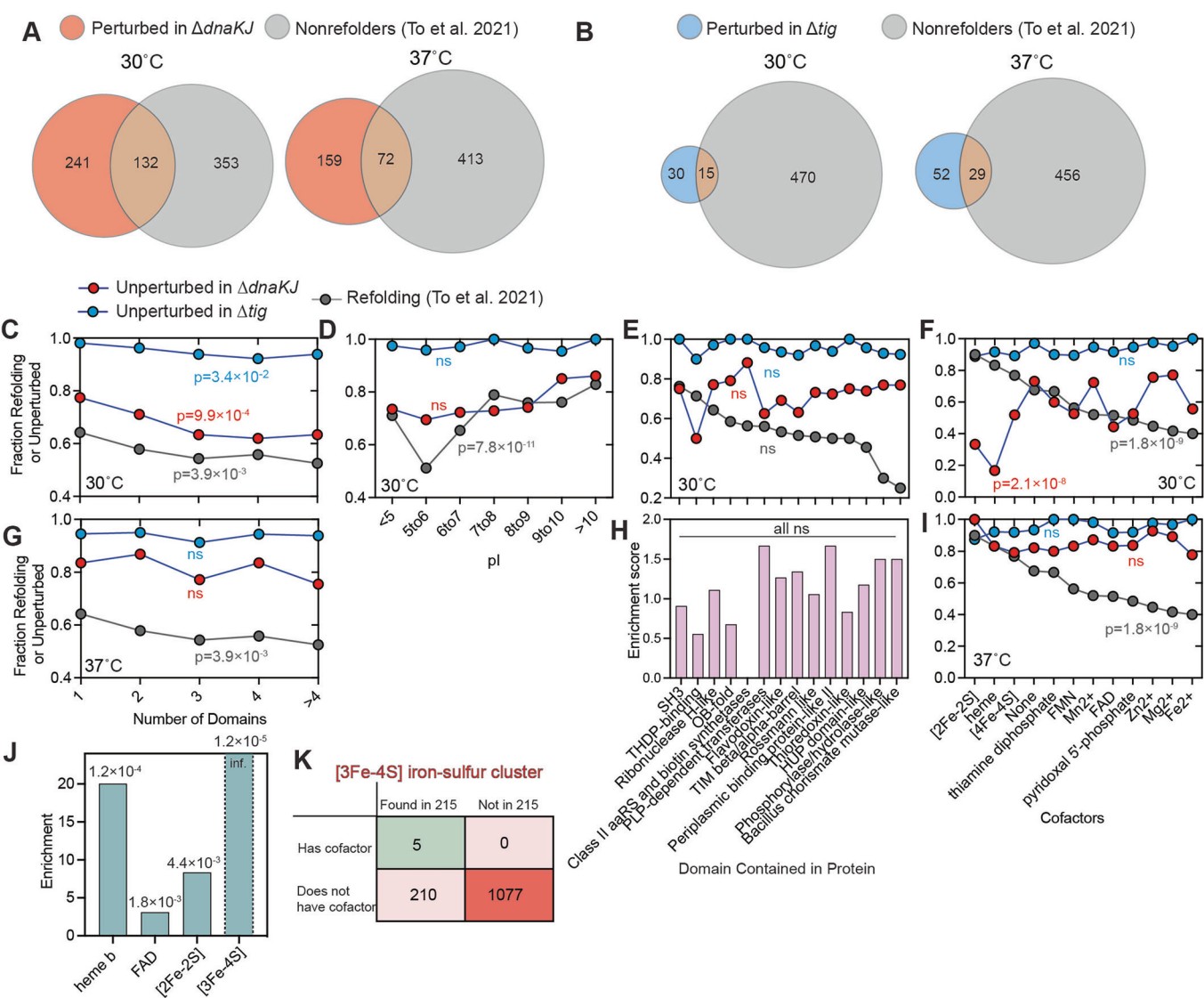

**Figure 2. Comparative analysis of protein refolding and de novo folding with chaperones knocked out at 30 °C and 37 °C.**

(**A, B**) Venn diagrams showing overlap between proteins that are structurally perturbed in Δ*dnaKJ* and Δ*tig* (at (**A**) 30 °C or (**B**) 37 °C) and/or nonrefolding from denaturant (according to To et al, 2021) (see Source Data for Fig. 2A,B). (**C–F**) Fraction of proteins that remain structurally unperturbed in Δ*dnaKJ* (red) or Δ*tig* (blue) at 30 °C compared to fraction of proteins that refold from denaturant according to To et al, 2021 (gray), grouped by (**C**) number of domains, (**D**) isoelectric point (pI), (**E**) whether it contains a domain of a given ECOD X-group fold type, (**F**) whether it hosts a given cofactor. *P* values are calculated according to the chi-square test. (**G, I**) Fraction of proteins that remain structurally unperturbed in Δ*dnaKJ* (red) or Δ*tig* (blue) at 37 °C compared to fraction of proteins that refold from denaturant according to To et al, 2021 (gray), grouped by (**G**) number of domains, (**I**) whether it hosts a given cofactor (see Source Data Fig. 2C–I). (**H**) Enrichments correspond to the frequency of proteins that contain a domain of a given ECOD X-group fold type within the group of 215 that are DnaKJ-dependent only at 30 °C relative to the overall frequency of proteins that contain that fold. *P* values calculated with Fisher's exact test are all non-significant. (**J**) Enrichments correspond to the frequency of proteins that host the given cofactor within the group of 215 that are DnaKJ-dependent only at 30 °C relative to the overall frequency of proteins that host the given cofactor. *P* values, written over the bars, are calculated using Fisher's exact test, using contingency tables such as the one shown in (**K**). (**K**) Contingency table showing an example of an enrichment analysis on the number of proteins that harbor a [3Fe-4S] iron–sulfur clusters (or not) and whether it was one of the 215 proteins that were structurally perturbed in Δ*dnaKJ* exclusively at 30 °C (or not). Source data are available online for this figure.

heme, and PLP (Fig. 2F, red trace). This finding can be rationalized. Covalently bound cofactors will not be detached during chemical denaturation, and once installed in the correct place, could serve as a "nucleus" to organize correct folding around them. On the other hand, during primary protein biogenesis, the correct attachment of

the covalently bound cofactor must be established, with states involving improperly attached cofactors acting as potential kinetic traps.

Strikingly, at 37 °C, the DnaKJ-dependence of certain cofactor-containing holoproteins virtually disappears (Fig. 2I), with

iron–sulfur cluster proteins now exhibiting virtually no structural perturbations in Δ*dnaKJ* and the dispersion in structure perturbation with respect to cofactors narrowing considerably (from a 60% spread to a 22% spread). We conjecture that misfolded conformations involving improperly attached cofactors might get kinetically trapped at 30 °C and require DnaK-stimulated ATP hydrolysis to be resolved; on the other hand, at 37 °C, the increased thermal energy could enable these misfolded states to resolve without chaperone assistance (He and Hiller, 2019).

Returning to the comparison between 30 °C and 37 °C (Fig. 1F), we find that indeed proteins with cofactors are enriched in the set of 215 proteins that were dependent on DnaKJ only at 30 °C (Fig. 2J,K). For instance, *all* the proteins with [3Fe–4S] clusters were found in this set of 215 proteins (Fig. 2K).

The relationship between DnaKJ-dependence and domain count also exhibited a temperature dependence reminiscent to what we observed with cofactors: Although at 30 °C, proteins with more domains appeared to be more reliant on DnaKJ to adopt their native structures (Fig. 2C), the effect disappears at 37 °C (Fig. 2G). A similar model to the one discussed could explain this finding. Misfolded conformations involving non-native interactions forming across domains might be metastable and kinetically trapped at 30 °C, requiring resolution by DnaK; however, these metastable states could resolve more spontaneously at elevated temperature.

We expected the 215 proteins that were dependent on DnaKJ at lower temperature to potentially involve topologically complex fold types (which may also be more prone to form kinetically trapped misfolded forms), but the data did not support this hypothesis (Fig. 2H). As an addendum, we note that, like at 30 °C, at 37 °C, DnaKJ-dependence did not vary meaningfully with pI (Fig. EV3A; Source Data Fig. EV3) or fold type (Fig. EV3B). At both temperatures, dependence went up for the proteins with the greatest molecular weight ( > 80 kDa, Fig. EV3C,D), though we could not detect any clear trend with oligomeric state (Fig. EV3E–H).

## Phosphoglycerate kinase is a nonrefolding protein which is not dependent on chaperones to fold in vivo

We have considered the glycolytic enzyme phosphoglycerate kinase (PGK) to be a model nonrefolding enzyme, because it possessed a very high number of structurally perturbed sites in LiP-MS experiments on refolded *E. coli* extracts (To et al, 2021), and we have shown it is prone to adopt metastable misfolded states that possess non-covalent lasso entanglements upon being compelled to refold from the denatured state (Jiang et al, 2025). Hence, we were surprised to see that despite very high levels of coverage in all four conditions we interrogated for this study (between 153 and 161 peptides matched), it was assessed as not structurally perturbed when either DnaKJ or Tig was knocked out, at 30 °C or 37 °C. Moreover, PGK is a chaperone-nonrefolder (To et al, 2022), meaning that it cannot refold from the denatured state even if GroEL/ES or DnaKJ is supplemented. These findings suggested PGK could be a "model" for a type of protein for which co-translational folding is the most important "chaperone" for its biogenesis; that is, it can fold efficiently on the ribosome (even in the absence of chaperone assistance), but its full-length form cannot refold from the denatured state (even with chaperone assistance).

To test this idea, we wanted to critically assess the "negative" result obtained from LiP-MS: Is it truly the case that PGK is structurally and functionally identical when it is biosynthesized in *E. coli* strains deficient in DnaKJ or trigger factor? To address this question, we used a scarless CRISPR-Cas9 method (Reisch and Prather, 2015) to install a StrepTag-II at the C-terminus of the *pgk* gene at its native chromosomal locus in the wild-type, Δ*dnaKJ*, and Δ*tig* backgrounds (Fig. 3A; Appendix Fig. S2A). These strains enable us to isolate PGK and characterize it in vitro when it is biosynthesized in distinct chaperone backgrounds without the additional potential artifacts that could accrue if PGK were over-expressed from a non-native plasmid system, which could be particularly serious in a chaperone knock-out strain, given that protein over-expression itself constitutes a form of stress (Schweder, 2002; Gill et al, 2000).

First, we verified by colony PCR the proper insertion of the epitope tag (Appendix Fig. S2B–E). Using Streptactin-XT resin and anion exchange, we were able to purify PGK from uninduced cells to homogeneity as assessed by SDS-PAGE (Appendix Fig. S3A–C). Intact mass spectrometry verified that the three samples of PGK were isolated with the expected mass for methionine loss and no additional post-translational modifications (Appendix Fig. S3D–F). We recorded the Michaelis-Menten kinetics from these PGK isolates by coupling the reverse reaction (ATP-dependent phosphorylation of 3-phosphoglycerate) to NADH depletion and measuring initial rates spectrophotometrically (Fig. 3B; Source Data Fig. 3B–E). The three PGK isolates displayed identical levels of enzymatic activity, as based on $k_{cat}$ and $K_M$ (Fig. 3C–E). Moreover, far-UV circular dichroism (CD) spectra of the three PGK isolates were superimposable (Fig. 3F; Source Data Fig. 3F–G), and consistent across replicates purified from separate biological cultures (Appendix Fig. S3G–I). Finally, we observed that regardless of chaperone background, PGK displayed identical melting temperatures as based on assessing its dichroism as a function of temperature (Fig. 3G). Hence, we conclude that a range of biophysical assays conducted on purified protein support the observation from LiP-MS that PGK is not structurally perturbed when DnaKJ or Tig is knocked out from *E. coli*.

To test if translation is truly sufficient to properly fold PGK without chaperone assistance, we carried out in vitro translation using purified components (Shimizu et al, 2001), and found that the protein purified in this way also showed similar kinetic parameters to those isolated from cells (Appendix Fig. S4D,E; Source Data Fig. S4D,E).

## Catalase establishes a positive control for LiP-MS on chaperone knock-outs

Protein structure is often related to protein function, however LiP-MS provides structural data in an abstract form which can call into question as to whether it provides a sufficiently resolved picture to produce useful functional predictions. This concern is particularly acute for the experiments presented here because unlike the large perturbation from unfolding proteins in 6 M guanidinium chloride, the structural effect from chaperone deletion is expected to be considerably more modest. For instance, at 37 °C, only 6.8% of all sequenced peptides showed significant changes in proteolytic susceptibility in Δ*dnaKJ*, (2.2% in Δ*tig*), implying that on average each protein only had 1.1 sites (0.4 sites in Δ*tig*) with evidence for

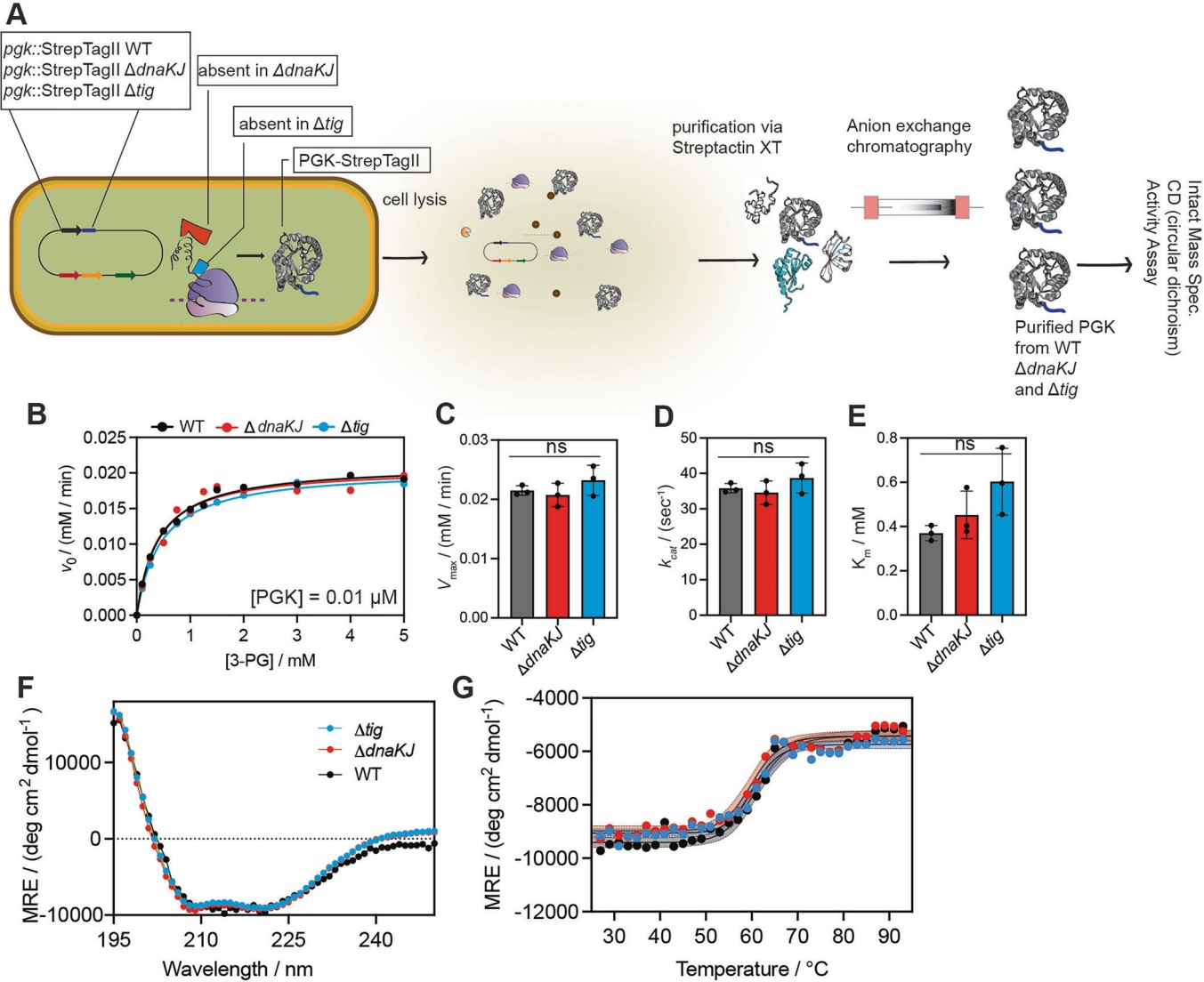

**Figure 3. Chromosomal tagging and biophysical characterization of PGK in wild-type and chaperone knockout *E. coli* strains.**

(A) Schematic of the CRISPR-based, scarless genomic tagging strategy used to insert a C-terminal Strep-tag II at the endogenous *pgk* locus in *E. coli* wild-type (WT), Δ*dnaKJ*, and Δ*tig* strains. Following native expression and folding in each strain background, PGK-Strep was purified via streptactin affinity chromatography and anion exchange. (B) Representative Michaelis-Menten curves showing initial rates as a function of initial substate concentration (3-phosphoglycerate) for PGK isolated from WT cells (black) or Δ*dnaKJ* (red) and Δ*tig* (blue) strains. (C–E) Maximal velocity (C), unimolecular rate constant ($k_{cat}$, D), and Michaelis constant ($K_M$, E) derived from fitting replicates ($n = 3$) of datasets such as the one shown in (B). Each dot represents the optimal value for $K_M$ and $k_{cat}$ based on a nonlinear fit of the initial rates at 8–10 substrate concentrations (see Source Data for Fig. 3B–E). Bars represent means, error bars represent standard deviations. *P* values calculated by one-way ANOVA with Bonferroni's multiple comparison test. ns not significant. (F) Far-UV CD spectra of PGK-Strep isolated from WT cells (black), Δ*dnaKJ* (red), and Δ*tig* (blue) strains. MRE, mean residue ellipticity. (G) Thermal denaturation curves monitored by CD at 220 nm of PGK-Strep isolated from WT cells (black), Δ*dnaKJ* (red), and Δ*tig* (blue) strains (see Source Data for Fig. 3F–G). Source data are available online for this figure.

structural alteration. To critically test if these smaller-scale structural changes could be confidently associated with function, we sought a positive control and focused on the primary *E. coli* catalase, KatG. KatG was modestly structurally perturbed in Δ*dnaKJ* at both 30 °C and 37 °C (with 5/49 and 2/66 significant cut-sites, respectively) but not perturbed in Δ*tig* at both 30 °C and 37 °C (with 1/48 and 1/73 significant cut-sites, respectively). Moreover, 2/5 of the sites at 30 °C (G180 and K201-K214) were

proximal to the active site porphyrin (2/2 of the sites at 37 °C: K335-R342 and R381-K388) and the functionally significant F and G helices (Li and Goodwin, 2004; Zámocký et al, 2001), according to an AlphaFold3 model (Fig. 4A), and we reasoned the activity of the enzyme could be monitored in lysate.

We applied a luminescence-based assay (Akimoto et al, 1990) to quantify catalase activity in clarified *E. coli* lysates from all three strains cultured at 30 °C. Specifically, catalase-containing extracts

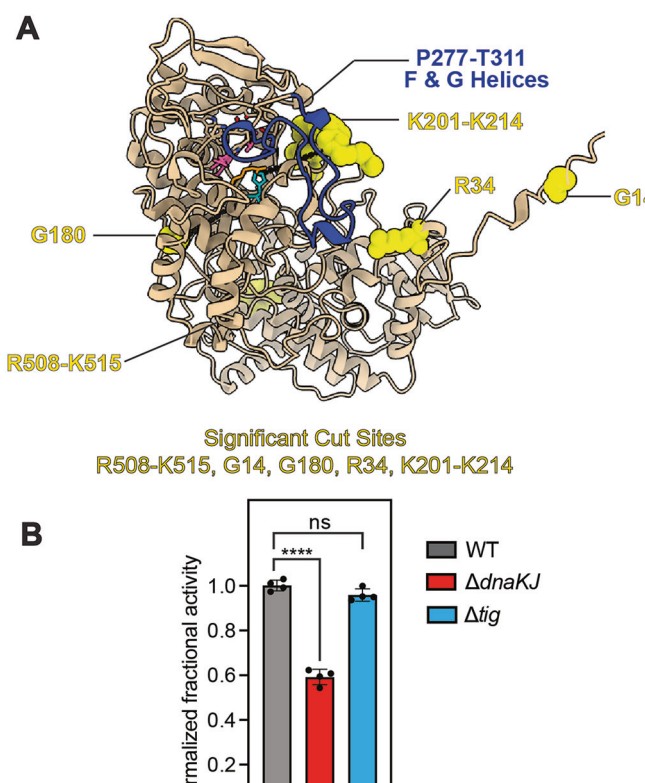

## A

**P277-T311**
**F & G Helices**

K201-K214

R34
G14

G180

R508-K515

**Significant Cut Sites**
R508-K515, G14, G180, R34, K201-K214

## B

**Figure 4. Catalase's structure and enzymatic activity are DnaKJ-dependent.**

**(A)** Predicted structure of *E. coli* KatG monomer generated using AlphaFold3. Significant proteolytic cut-sites identified by LiP-MS in the Δ*dnaKJ* strain at 30 °C are highlighted as yellow spheres. Heme is colored pink, active site residues H106 and R102 are colored cyan and orange, and F-G helices (P277-T311) are colored blue. **(B)** Catalase activity assay performed using clarified lysates from WT, Δ*dnaKJ*, and Δ*tig* strains ($n = 3$) at 0.4 mg/mL total protein concentration (see Source Data for Fig. 4B). Note KatG abundance is similar across strains (cf. Fig. EV4). Bars represent means, error bars represent standard deviations. *P* values calculated by one-way ANOVA with Bonferroni's multiple comparison test. ns not significant; ****$P < 0.0001$. Source data are available online for this figure.

were combined with hydrogen peroxide ($H_2O_2$) and incubated for 1 min, and the remaining $H_2O_2$ was quantified by converting to a chemiluminescent signal with horseradish peroxidase and luminol (Fig. EV4A). Luminescence decay curves recorded from lysates at three different concentrations (0.2, 0.3, and 0.4 mg/mL) revealed consistently reduced $H_2O_2$ degradation from Δ*dnaKJ* extracts compared to those from wild-type and Δ*tig* (Fig. EV4B,C). To ensure that differences in total activity were not driven by variations in KatG expression, we quantified protein abundance using MaxLFQ values from label-free proteomics. Interestingly, KatG levels were slightly elevated in Δ*dnaKJ* relative to WT and Δ*tig* (Fig. EV4D), ruling out reduced expression as the source for lower total enzymatic activity.

Normalized fractional activity was calculated (Figs. 4B and EV4E,F), and it was found that at all lysate concentrations tested (0.2, 0.3, and 0.4 mg/mL), Δ*dnaKJ* extracts showed reduced

specific catalase activity, while the activity of Δ*tig* extracts were indistinguishable from wild-type. Together, these results show that catalase activity is impaired due to structural perturbations that occur when it is biosynthesized in cells without DnaKJ. Given that KatG was structurally perturbed at multiple sites in Δ*dnaKJ* (Fig. 4A), and some of these perturbations coincide with regions near the active site, we suggest that DnaKJ is essential for the proper folding of active KatG under native growth conditions, and that the structural data from LiP-MS can make reliable predictions about protein function.

## Comparisons of chaperone dependence in vivo and in vitro

In the cell, molecular chaperones play diverse functions in primary protein biogenesis (co-translationally or immediately post-translationally) and in stress response (such as repairing misfolded conformations or preventing aggregation). Although unrelated to their main functions in vivo, chaperones also assist the refolding of proteins that have been fully unfolded by denaturant in vitro (Huang et al, 2000; Clerico et al, 2019; Diamant et al, 2001; Goloubinoff et al, 1989b). We were intrigued to make comparisons between these functions by comparing the LiP-MS data we obtained here to interrogate DnaKJ-dependency during protein folding in vivo to previous LiP-MS experiments in which chaperones were supplemented to in vitro refolding reactions comprising of the *E. coli* proteome initially denatured in 6 M guanidinium chloride (To et al, 2022).

First, we reanalyzed the data from To et al. using FragPipe and FLiPPR in order to harmonize the earlier work with our current data processing workflow including FDR-control to assign proteins a status of "structurally (un-)perturbed" or "(non)refoldable" (Data ref: To et al, 2022; Source Data for Fig. 5). When cross-correlating refolding experiments without chaperones (whose overall refolding rate was 58.2%, 603 out of 1029 proteins) to refolding experiments with DnaKJ and GrpE (whose overall refolding rate was 77.5%, 564 out of 728 proteins), we found that 701 could be confidently assessed in both, of which 216 were DnaK-dependent refolders (that is, they refolded with DnaKJ but not without; note this set would *not* include proteins that require DnaKJ along with other chaperones; Fig. 5A). We next compared that list of 216 DnaKJ-dependent refolders to the 373 DnaKJ-dependent proteins in the current study (which were structurally perturbed in Δ*dnaKJ* cells). Of the resulting 525 proteins, only 64 (12.2%) were shared in common (Fig. 5B). When the same comparison is done using the Δ*dnaKJ* dataset at 37 °C (where DnaKJ is less required), the overlap decreases further to 10.4% (Fig. 5C). In other words, the data suggest that the proteins which require DnaKJ in vivo during primary biogenesis are generally unrelated to the proteins which require DnaKJ in vitro during total refolding. The only obvious reason why this could be is that in vivo DnaKJ primarily acts co-translationally and interacts with a distinct set of intermediates, which are partially unfolded/misfolded rather than fully unfolded.

To understand this logic, consider the extreme possibility in which proteins did not fold co-translationally, and emerged from the ribosome in a denatured state, whereupon DnaKJ would then facilitate their folding (Fig. 5D). If this were true, then the overlap between DnaKJ-dependence in vivo and during in vitro refolding

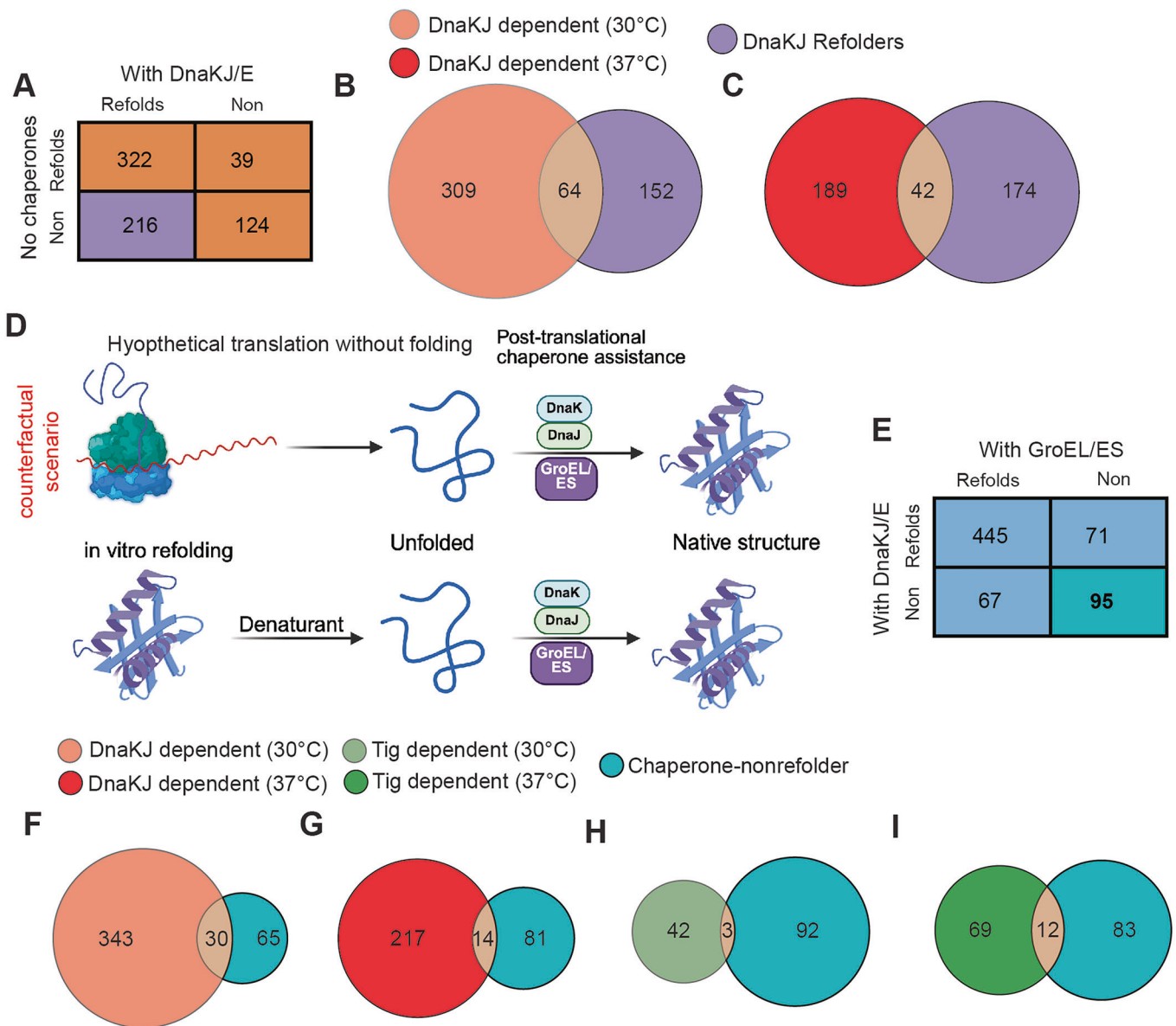

**Figure 5. Comparison of chaperone dependencies in vivo and in vitro.**

(A) Contingency table from chaperone refolding experiments (To et al, 2022) that defines which proteins are categorized as DnaKJ-dependent refolders (see Source Data for Fig. 5A). (B) Venn diagram showing overlap between in vitro DnaKJ-dependent refolders ($n = 216$) and in vivo DnaKJ-dependent substrates ($n = 373$) identified by structural perturbation in $\Delta dnaKJ$ cells at 30 °C. (C) Same comparison as in (B), but using in vivo $\Delta dnaKJ$ LiP-MS data at 37 °C (see Source Data for Fig. 5B,C). (D) Cartoon illustrating a hypothetical model where no co-translational folding occurs, such that nascent proteins emerge in an unfolded state and require DnaKJ for folding post-translationally. (E) Contingency table from chaperone refolding experiments (To et al, 2022) that defines which proteins are categorized as chaperone-nonrefolders (see Source Data for Fig. 5E). (F–I) Venn Diagrams showing the overlap between the list of 95 chaperone-nonrefolders and proteins which are DnaKJ-dependent (F, G) or Tig-dependent (H, I) in vivo at either 30 °C (F, H) or 37 °C (G, I) (see Source Data for Fig. 5F,G and Fig. 5H,I). Source data are available online for this figure.

would be perfect. In reality, the overlap is as low as it is to GroEL-dependent refolders (12.1%, Fig. EV5A–C).

To highlight another way in which in vivo primary biogenesis and in vitro refolding differ, we consider the case of PGK. Earlier, we referred to PGK as a chaperone-nonrefolder. This status is assigned to 95 proteins (Fig. 5E), based on a cross-correlation analysis comparing refolding experiments with DnaKJ to those with GroEL/ES (whose overall refolding rate was 81.5%, 780 out of 957

proteins). We found that 678 proteins could be confidently assessed in both, of which 95 were assigned nonrefolding statuses under both chaperone conditions. We note that this figure is lower than the 105 reported in (To et al, 2022) due to the protein-wise FDR correction applied here. In our earlier work, we conjectured that a protein could be chaperone-nonrefolding if it is an "obligate co-translational folder." If indeed co-translational folding is *the most* important factor (or even the only factor) necessary for properly

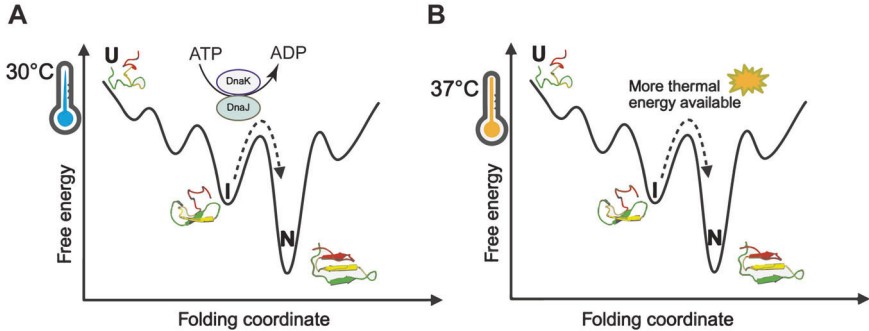

**Figure 6. Increased dependence on DnaKJ at low temperature reflects limited thermal energy to escape kinetic traps during folding.**

Energy landscape diagrams showing putative folding pathways at 30 °C (**A**) and 37 °C (**B**).

folding these proteins, then we might expect chaperone-nonrefolders to not demonstrate much dependence on chaperones in vivo. That expectation is the case. When we compared the list of chaperone-nonrefolders to those that were structurally perturbed in Δ*dnaKJ* or in Δ*tig* (Fig. 5F-I), we found that there was very low overlap (between 2 and 7%). In other words, PGK is representative of the chaperone-nonrefolder class as a whole, in its not being structurally perturbed when chaperones are knocked out in vivo. Ostensibly, co-translational folding plays the most significant role in enabling it to successfully locate its native conformation.

## Discussion

By knocking out two of the three central chaperones in *E. coli*, and using LiP-MS to record the structural perturbations incurred by the proteome in these genetic backgrounds with impaired protein folding machinery, we were able to make three interesting observations. Firstly, our study provides support to the view that DnaK (Hsp70) plays a central role in the primary biogenesis of many *E. coli* proteins during normal growth conditions, whilst trigger factor is more dispensable. It is possible that some proteins fold more slowly or with lower yield without it, but ultimately native structures for much of the proteome can be achieved without it. While GroEL is another critical chaperone in *E. coli*, it is essential and cannot be knocked out, which would make studying the structural consequences of its removal in living cells more challenging. We observe upregulation of GroEL in Δ*dnaKJ* cells, raising the possibility that some DnaKJ clients may be rerouted to the GroEL/ES system in the absence of DnaKJ. It is likely that this shift helps mitigate the extent of structural perturbations in Δ*dnaKJ*.

Second, our results reveal a surprising feature about the temperature dependence of DnaK: outside of stress conditions, it is more important at lower temperature likely because it can couple ATP hydrolysis to unfold metastable misfolded states (Fig. 6A,B). This effect is most pronounced for proteins with covalently bound cofactors. In some ways, this observation is reminiscent of RNA maintenance in *E. coli*, as RNA helicases (i.e., DeaD) are overexpressed at lower temperatures where RNA would be more prone to get trapped in metastable states (Phadtare and Severinov, 2010). Thirdly and perhaps most significantly, this study—in

comparison to a previous one of ours on chaperone assistance during in vitro refolding (To et al, 2022)—highlights the very distinct role DnaKJ plays during primary biogenesis and in vitro refolding and points to co-translational folding itself sometimes being the most important chaperone.

Because co-translational folding can simplify the folding problem significantly (by enabling proteins to fold vectorially and modularly), folding problems that can "seem hard" based on in vitro assays can be relatively easier for cells (Thommen et al, 2017; Komar, 2009; Jacobs and Shakhnovich, 2017; Liutkute et al, 2020; Kramer et al, 2019). Two interesting examples of this can be gathered from our observations concerning fold types and cofactors. Certain fold-types do not refold efficiently, particularly larger domains with α/β content and more complex topology (Fig. 2C); on the other hand, it does not appear that different fold-types are generally more DnaKJ-dependent or Tig-dependent during folding in vivo, with the interesting exception of the THDP-binding domain (cf. Fig. 2E). This may be because the "more complex" domains specifically evolved to fold efficiently on the ribosome, and because they tend to be more stable than smaller domains, they may not have been under selective pressure to refold. Likewise, certain holoproteins are complicated to refold, especially if they host cofactors that would come detached during unfolding and require the formation of a specific coordination environment to reinsert during refolding (Fig. 2C). However, at 37 °C, cofactor-containing proteins do not generally require chaperones like Tig or DnaKJ, and many form with greater likelihoods than proteins which do not contain cofactors. Hence, co-translational folding may also be an important factor to assist the biosynthesis of cofactor-containing proteins.

More broadly, the LiP-MS data reported in this study—supported by our in vitro characterization of PGK and activity assays on KatG—highlight the usefulness of structural proteomics to elucidate chaperone mechanisms holistically and on native clients. By focusing on the actual structural outcome of protein biogenesis, rather than on chaperone binding (which can be measured by co-precipitation mass spectrometry (Kerner et al, 2005; Calloni et al, 2012) and by selective ribosome profiling (SeRP, Oh et al, 2011; Shiber et al, 2018; Stein et al, 2019; Galmozzi et al, 2025)), we can see that chaperones sometimes are involved in folding processes without being required to achieve natively-folded structures. Although our experiments do not directly report on the

structure of nascent chains or newly synthesized proteins, we can compare our results to a recent SeRP study that assessed which nascent proteins engage *E. coli* chaperones co-translationally (Data ref: Galmozzi et al, 2025). We find that proteins which become structurally perturbed in Δ*dnaKJ* indeed interact more with DnaKJ as nascent chains ($P < 3 \times 10^{-4}$ by Kolmogorov–Smirnov test; Fig. EV5D,E, Source Data for Fig. EV5). This observation supports the view that under non-stress conditions, DnaKJ supports co-translational folding, and its absence during protein synthesis results in structural changes that persist following ejection from the ribosome. By contrast, we did not find that proteins which become structurally perturbed in Δ*tig* possessed greater interaction with trigger factor as nascent chains (Fig. EV5F,G). This result could either be simply due to less statistical power from having fewer examples, or alternatively could be construed as showing that the trigger factor binds to many nascent proteins but is not essential for the acquisition of native structure.

Finally, taken together with our previous investigations of the global refolding reactions with and without chaperones, this study provides a cautionary tale to not generalize the functions played by chaperones during in vitro refolding (wherein they interact with conformations that populate from a fully unfolded form) to the functions they play during primary biogenesis in vivo. This finding appears to be in keeping with recent advances in the chaperone mechanism field, where new approaches to study these remarkable machines in their native context are emerging (Thommen et al, 2017), and where a new consensus is building of the key role they play during co-translational folding (Roeselová et al, 2024b; Wales et al, 2024; Balchin et al, 2020; Galmozzi et al, 2025; Oh et al, 2011).

# Methods

### Reagents and tools table

| Reagent/resource | Reference or source | Identifier or catalog number |
|---|---|---|
| **Bacterial strains (E. coli)** | | |
| WT (strain MC4100) Δ*dnaKdnaJ*::Kan^R^(ΔKJ) Δ*tig*::Cm^R^(ΔT) | Gifts from Hartl Lab | |
| **Recombinant DNA/Oligonucleotides** | | |
| Primers | This paper | Table S1 |
| pCas9cr4 | Reisch and Prather, 2015 | |
| pKDsgRNA_p15 | Reisch and Prather, 2015 | |
| **Antibodies** | | |
| 6x-His Tag Monoclonal Antibody (HIS.H8) | ThermoFisher Scientific | MA1-21315 |
| Goat anti-Mouse IgG (H + L) Secondary Antibody, HRP | Invitrogen | 31430 |
| **Chemicals, enzymes, and other Reagents** | | |
| Adenosine 5′-triphosphate magnesium salt | Millipore Sigma | A9187 |
| Ammonium Bicarbonate | Acros Organics | 393219950 |
| Dithiothreitol (DTT) | Sigma-Aldrich | D06632 |
| Glycerol | Fisher Scientific | G33-1 |

| Reagent/resource | Reference or source | Identifier or catalog number |
|---|---|---|
| Iodoacetamide (IAA) | Acros Organics | 50-01-1 |
| KCl | Fisher Scientific | BP366-1 |
| MgCl$_2$ | Fisher Scientific | BP214-500 |
| NaCl | Fisher Scientific | BP358-212 |
| NADH | Sigma-Aldrich | 10128023001 |
| D-3-Phosphoglyceric acid disodium salt | Sigma-Aldrich | P8877 |
| Tris-Base | Sigma-Aldrich | RD008 |
| GAPDH | Millipore Sigma | G2267 |
| Tris-HCl | Sigma | RD009 |
| Tryptone | Fisher | BP1421-2 |
| Urea | Fisher | U17-212 |
| Yeast Extract | Fisher | BP1422-500 |
| Proteinase K | Fisher Scientific | BP1700 |
| PMSF | Millipore Sigma | 10837091001 |
| Pepstatin A | ThermoFisher Scientific | 78436 |
| E-64 | ThermoFisher Scientific | 78434 |
| Bestatin | ThermoFisher Scientific | 78433 |
| DNase I from Bovine Pancreas | Sigma-Aldrich | 31136 |
| Rapid Gold BCA Assay (Pierce) | ThermoFisher Scientific | A53225 |
| Mineral Oil | | |
| Trypsin-ultra Mass Spectrometry Grade | New England Biolabs | P8101S |
| Purefrex 2.1 | Cosmobiousa | GFK-PF213-0.25-EX |
| RNase Inhibitor murine | NEB | M0314S |
| HisPur Ni-NTA Magnetic beads | ThermoFisher Scientific | 88831 |
| Streptactin XT 4 Flow High Capacity resin | Iba lifesciences | 2-5030-002 |
| Buffer BXT | Iba lifesciences | 2-1042-025 |
| Buffer W | Iba lifesciences | 2-1003-100 |
| Imidazole | ThermoFisher Scientific | A10221.22 |
| Hydrogen Peroxide | Millipore Sigma | HX0636-1 |
| Luminol (SuperSignal™ West Femto Maximum Sensitivity Substrate) | ThermoFisher Scientific | 34096 |
| Streptactin-HRP | Iba lifesciences | 2-1502-001 |
| Acetonitrile (HPLC Grade) | Fisher Scientific | A955-4 |
| Formic acid (Optima LC/MS Grade) | Fisher Scientific | A117-05AMP |
| Trifluoroacetic acid (TFA) (Optima LC/MS Grade) | Fisher Scientific | A116-50 |
| Water (Optima LC-MS Grade) | Fisher Scientific | W6500 |
| **Instrumentation** | | |
| ¼- inch Probe | QSonica | N/A |
| 1/8-inch Probe | QSonica | N/A |
| Eppendorf Centrifuge | Eppendorf | N/A |

| Reagent/resource | Reference or source | Identifier or catalog number |
|---|---|---|
| pH meter | Mettler-Toledo | N/A |
| Plate Reader (iD3) | Molecular Devices | N/A |
| Probe Sonicator | QSonica | N/A |
| Q-Exactive HF-X Orbitrap Mass Spectrometer | Thermo Scientific | N/A |
| SPEX™ Sample Prep Dual Freezer/Mill | SPEX | N/A |
| SW55 Ti Rotor | Beckman Coulter | N/A |
| UltiMate3000 UHPLC | Thermo Scientific | N/A |
| UltraCentrifuge (Optima XL-A) | Beckman Coulter | N/A |
| Circular Dichroism Spectrometer | Aviv 420 CD | N/A |
| Akta Go | Cytiva | N/A |
| **Software and algorithms** | | |
| Xcalibur | Thermo Scientific | |
| FragPipe | https://fragpipe.nesvilab.org/ | |
| Flippr | (Manriquez-Sandoval et al, 2024) | |
| GraphPad Prism | https://graphpad.com/ | |
| Adobe Illustrator | https://www.adobe.com/products/illustrator.html | |
| ShinyGo 0.82 | https://bioinformatics.sdstate.edu/go/ | |
| BioRender | https://app.biorender.com/ | |
| **Other** | | |
| 3 ml Konical tubes | Beckman Coulter | |
| JL20 Tubes | Beckman Coulter | |
| Sep Pak C18 1cc 50 mg Resin | Waters | 186000308 |

## Methods and protocols

Experiments in this study were conducted without blinding.

### Culture and lysis of Escherichia coli (MC4100)

Saturated overnight cultures of *E. coli* cells, strain MC4100 WT, $\Delta dnaKJ$, $\Delta tig$ (gift from Hartl Lab) were used to inoculate $3 \times 100$ mL (biological triplicates) cultures in LB Media in 250-mL flasks at a starting $OD_{600}$ of 0.05. Cells were grown at 37 °C and 30 °C with agitation (220 rpm) to a final OD 600 of 0.8, followed by centrifugation at $4000 \times g$ for 15 min at 4 °C. Supernatants were removed, and cell pellets were stored overnight at −80 °C until further use.

### Preparation of normalized lysates

Frozen cell pellets were thawed on ice and resuspended in 2.0 mL of native buffer (20 mM Tris-HCl pH 8.2, 100 mM KCl, 2 mM MgCl$_2$) with the addition DNase I (NEB M0303S) to a final concentration of 0.1 mg/ml and protease inhibitors (500 µM PMSF (Thermo Scientific 36978); 15 µM E-64 (Millipore Sigma E3132); 50 µM Bestatin (Millipore Sigma B8385)).

Resuspended cells were flash-frozen by slow dripping over liquid nitrogen and then cryogenically pulverized with a freezer mill (SPEX Sample Prep) over 8 cycles consisting of 1 min of grinding, 9 Hz, and 1 min of cooling. Pulverized lysates were transferred to a 50 mL centrifuge tube and thawed at room temperature for 20 min. Clarified lysates were then transferred into 3 mL Beckman Coulter Konical tubes (Beckman Coulter C14307) in preparation for ultracentrifugation at 33,300 rpm at 4 °C for 90 min using a SW55-Ti rotor without a sucrose cushion to deplete ribosomes. These clarified supernatants were then transferred into new 1.5-mL microcentrifuge tubes, and protein concentrations were determined using a bicinchoninic acid assay (Rapid Gold BCA Assay, Pierce) in a clear 96-well plate (Corning Falcon 353075) with a plate reader (Molecular Devices iD3). Using the results from the BCA Assay, the clarified cellular lysates were normalized to a protein concentration of 3.3 mg mL$^{-1}$ using the same lysis buffer (20 mM Tris pH 8.2, 100 mM KCl, 2 mM MgCl$_2$).

### Preparation of LiP and control (trypsin-only) samples

The LiP and control samples were prepared by diluting lysates with native dilution buffer (20 mM Tris pH 8.2, 100 mM KCl, 2 mM MgCl$_2$) such that upon dilution, the final concentrations of all components were 20 mM Tris pH 8.2, 100 mM KCl, 2 mM MgCl$_2$, and a protein concentration of 0.23 mg mL$^{-1}$.

### Limited proteolysis and mass spectrometry sample preparation

A 2 µL portion of a Proteinase K (PK) stock (prepared as a 0.23 mg mL$^{-1}$ PK in a 1:1 mixture of native lysis buffer and 20% glycerol, stored at −20 °C and thawed at most only once) was added to a fresh 1.5-mL microfuge tube. In triplicate, 200 µL of LiP sample was then added to the PK-containing microfuge tube and rapidly mixed by pipetting (1:100 enzyme:substrate mass ratio). Samples were incubated for exactly 1 min at 25 °C before being transferred to a mineral oil bath, preequilibrated at 100 °C, and incubated for 5 min to quench PK activity. Boiled samples were then transferred to a fresh 2-mL microfuge tube containing 200 mg urea and 85 µL of native buffer such that the final urea concentration was 8 M and the final volume was 415 µL. This method generates the limited proteolysis sample (LiP) protein samples. For samples designated as controls (Trypsin-Only), the same procedure was used as above, except PK was not added. In total, 18 samples were prepared for this experiment for each temperature 30 °C and 37 °C, each done in biological triplicates. All samples then received 6.24 µL of freshly prepared 500 mM DTT (final concentration 10 mM DTT) to reduce disulfide bonds and were incubated for 30 min at 37 °C with agitation at 700 rpm followed by 17 µL of freshly prepared 750 mM IAA (final concentration 40 mM IAA) and incubated for 45 min in the dark at 25 °C to alkylate reduced cysteines. Samples were then diluted with 1005 µL of 100 mM ammonium bicarbonate, followed by 4 µL of 0.116 mg mL$^{-1}$ of trypsin-ultra (NEB P8101) and incubated at 25 °C for 16–24 h at 25 °C. Trypsin was then quenched with trifluoroacetic acid (1% v/v final concentration) prior to desalting with Sep-Pak C18 1 cm$^3$ cartridges (Waters).

Cartridges were first conditioned ($2 \times 1$ mL 80% ACN, 0.5% TFA) and equilibrated ($4 \times 1$ mL 0.5% TFA) before samples were

slowly loaded under a weak vacuum. The columns were then washed ($4 \times 1\,mL$ 0.5% TFA), and peptides were eluted by the addition of 1 mL of elution buffer (80% ACN, 0.5% TFA). During elution, vacuum cartridges were suspended above 15 mL conical tubes, placed in a swing-bucket rotor (Eppendorf 5910 R), and spun for 5 min at $300 \times g$. Eluted peptides were transferred from Falcon tubes back into new 1.5 mL microcentrifuge tubes and dried using a Vacufuge Plus (Eppendorf). Dried peptides were stored at $-80\,°C$ until analysis. For analysis, samples were vigorously resuspended in 0.1% FA in Optima water (ThermoFisher) to a final concentration of $1\,mg\,mL^{-1}$.

### Mass spectrometry acquisition

Chromatographic separation of digests was carried out on a Thermo UltiMate3000 UHPLC system with an AcclaimPepmap RSLC, C18, $75\,\mu m \times 25\,cm$, $2\,\mu m$, $100\,Å$ column. Approximately $1.5\,\mu g$ of protein was injected onto the column. The column temperature was maintained at $40\,°C$, and the flow rate was set to $0.300\,\mu L\,min^{-1}$ for the duration of the run. Solvent A (0.1%FA) and Solvent B (0.1% FA in ACN) were used as the chromatography solvents. The samples were run through the UHPLC System as follows: peptides were allowed to accumulate onto the trap column (AcclaimPepMap 100, C18, $75\,\mu m \times 2\,cm$, $3\,\mu m$, $100\,Å$ column) for 10 min (during which the column was held at 2% Solvent B). The peptides were resolved by switching the trap column to be in-line with the separating column, quickly increasing the gradient to 5% B over 5 min and then applying a 95 min linear gradient from 5% B to 25% B. Subsequently, the gradient was increased from 25% B to 40% B over 25 min and then increased again from 40% B to 90% B over 5 min. The column was then cleaned with a sawtooth gradient to purge residual peptides between runs in a sequence.

A Thermo Q-Exactive HF-X Orbitrap mass spectrometer was used to analyze protein digests. A full MS scan in positive ion mode was followed by 20 data-dependent MS scans. The full MS scan was collected using a resolution of 120,000 (@ $m/z$ 200), an AGC target of 3E6, a maximum injection time of 64 ms, and a scan range from 350 to 1500 $m/z$. The data-dependent scans were collected with a solution of 15,000 (@ $m/z$ 200), an AGC target of 1E5, a minimum AGC target of 8E3, a maximum injection time of 55 ms, and an isolation window of 1.4 $m/z$ units. To dissociate precursors prior to their reanalysis by MS2, peptides were subjected to an HCD of 28% normalized collision energies. Fragments with charges of 1, 6, 7, or higher and unassigned were excluded from analysis, and a dynamic exclusion window of 30.0 s was used for the data-dependent scans.

### Primary analysis of MS data in FragPipe and FLiPPR

FragPipe v20.0, along with MSFragger v3.8, IonQuant v1.9.8, and Philosopher v5.0, were used to analyze raw mass spectra with label-free quantification with match between runs enabled. Default settings were used except that the precursor mass tolerance was changed to $-10$ to 10 ppm, the peptide digest pattern was set to semi-enzymatic, methionine oxidation and N-terminal acetylation were set as dynamic modification, and cysteine carbamidomethylation was set as a static modification. Two label-free quantifications (LFQs) were carried out FragPipe, one comparing the replicates of the 3 strains' LiP samples (9 raw files total), and a second comparing the replicates of the 3 strains' trypsin-only control samples (9 raw files total). The ion files of these LFQs were then passed to FLiPPR v0.0.7 to perform imputation, FDR correction,

normalization to protein abundance, and ion to cut-site merging. For this study, we focused on the cut-site counts. In general, proteins were considered for analyses if two or more cut-sites were assessed, and proteins were labeled nonrefoldable or structurally perturbed if they possessed two or more cut-sites with a statistically significant difference in a knock-out strain compared to WT. Output files from FLiPPR were then further processed in MS Excel and GraphPad Prism to create summaries and graphical representations of all datasets.

### Metadata generation and annotation mapping

To investigate trends in protein refoldability and structural perturbation across *E. coli* proteomes, we compiled a comprehensive metadata file primarily using annotations from the EcoCyc database (https://ecocyc.org). This included curated information for *E. coli* K-12 on gene function, cellular localization, subunit composition, essentiality, copy number, cofactors, and molecular weight. Since our experiments used the *E. coli* MC4100 strain, metadata from K-12 was mapped to MC4100 using gene names and known synonyms. A custom Python script automated this mapping and ensured consistent formatting and indexing by gene symbol. Additional annotations were incorporated from the ECOD database (http://prodata.swmed.edu/ecod/) to assign X-groups to each structural domain. Domain annotations were converted from UniProt IDs to gene symbols, and the number and types of domains per protein were recorded. Isoelectric point (pI) values were extracted from the isoelectric point database, and the average pI across all algorithms was calculated for each protein. The finalized metadata file was merged with protein-level summaries from our FLiPPR analyses at 30 °C and 37 °C using gene symbols as unique identifiers. This integration facilitated direct comparison between structural proteomics results and protein characteristics.

### Structural perturbation criteria and classification

Proteins were classified as "structurally perturbed" under a given condition if they had two or more significant cut-sites (effect-size >2, and Benjamini–Hochberg adjusted $P < 0.05$). Proteins were classified as structurally unperturbed if they had zero or one significant cut-sites, though they still needed to have two or more valid quantified cut-sites. Proteins with only one cut-site quantified were excluded from subsequent analyses. This classification scheme was applied consistently to define chaperone dependence in vivo or refoldability in vitro. Protein-level contingency tables were constructed to quantify overlap in proteins' status (e.g., perturbed vs. unperturbed, refolder vs. non-refolder) across two conditions. The outcomes for a protein in two conditions were pooled by using the gene symbol as the common identifier. Statistical enrichment of various categories (e.g., pI, molecular weight, number of domains, or domain architecture) among perturbed or refolding-sensitive proteins was evaluated using the chi-square tests or Fisher's exact tests where applicable (in the case of a $2 \times 2$ table).

### Reanalysis of in vitro refolding (To et al, 2022)

We reanalyzed LiP-MS data from To et al (To et al, 2022) using three independent label-free quantification (LFQ) runs in FragPipe, corresponding to refolding time courses with and without molecular chaperones. Each LFQ involved 12 raw files: three replicates for the native condition and three each for refolding at 1 min, 5 min, and 2 h. This process was performed separately for

samples containing: No chaperones, GroEL/ES, and DnaKJ/GrpE. Ion quantification outputs were processed using FLiPPR v0.0.7, which performed missing value imputation, FDR correction, and cut-site merging. No normalization for protein abundance was applied. For this study, the 5-min refolding time was used to classify proteins as "refolders" or "non-refolders," using the same criteria outlined above (≥ 2 significant cut-sites).

### Contingency tables and Venn diagram analysis

To compare refolding outcomes between in vivo chaperone dependence and in vitro refoldability, we constructed two-way contingency tables by overlapping reanalyzed data from To et al with our in vivo datasets. For example, Fig. 5A compares refolding classification in the presence vs. absence of DnaKJ. Figure 5E compares refolding outcomes under DnaKJ vs. GroEL/ES supplementation. Proteins were grouped as "refolder" or "non-refolder" based on their refolding performance in the presence of each chaperone. Venn diagrams (Figs. 5B,C,F–I and EV5A–C) visualize overlaps between DnaKJ-dependent proteins identified in vivo (at both 30 °C and 37 °C) and those refoldable in vitro with either DnaKJ or GroEL/ES. Binary overlaps were computed using gene symbols as identifiers. Protein sets for these diagrams were defined using the same perturbation/refoldability criteria as above. Venn diagrams were generated using Python's matplotlib and matplotlib_venn libraries, which calculated and visualized set intersections across two- or three-way comparisons.

### Scarless genomic tagging of PGK with StrepTag II using CRISPR-Cas9

A C-terminal StrepTag II was introduced into the chromosomal *pgk* locus in *Escherichia coli* strains MC4100 WT, Δ*dnaKJ*, and Δ*tig* using a scarless CRISPR-Cas9 genome editing approach as previously described by Reisch and Prather, 2015. Plasmids used in this system included pCas9cr4 (Addgene #62655) for expression of Cas9 and λ-Red recombinase, and pKDsg-pgk (Addgene #62654) or pKDsg-p15 (Addgene #62656) for sgRNA expression targeting the 3′ end of *pgk*. All plasmids were transformed into *E. coli* WT, Δ*dnaKJ*, and Δ*tig* via electroporation and maintained under chloramphenicol (34 µg/mL) and spectinomycin (50 µg/mL) selection at 30 °C. Oligonucleotides used in these manipulations are listed in Appendix Table S1.

To induce λ-Red expression, overnight cultures of strains harboring both plasmids were subcultured into SOB medium containing antibiotics and grown at 30 °C until $OD_{600}$ reached 0.4–0.5. L-arabinose was added to a final concentration of 50 mM, and cultures were incubated for 20 min. Cells were chilled on ice, pelleted at 2000×g for 15 min at 4 °C, and made electrocompetent using a glycerol–mannitol step gradient. The cell pellet was resuspended in 100 µL of cold glycerol–mannitol solution.

Approximately 150 ng of PCR-amplified donor DNA (gBlock from IDT containing a C-terminal StrepTag II flanked by 100 bp homology arms) was added to 100 µL of competent cells and electroporated in a 1 mm cuvette at 1.8 kV using an Eppendorf Eporator. The electroporated cells were immediately recovered in 1 mL of SOC at 30 °C with shaking for 1.5 h, then plated on LB agar supplemented with chloramphenicol, spectinomycin, and 100 ng/mL anhydrotetracycline (aTc). Plates were incubated overnight at 30 °C.

Candidate colonies were screened using colony PCR. Individual colonies were suspended in 100 µL of deionized water, and 10 µL was used as template in a 25 µL PCR reaction using PGK-specific

primers (5 µM each), OneTaq 2× Master Mix (NEB), and the following thermocycling conditions: initial denaturation at 94 °C for 30 s, followed by 30 cycles of 94 °C for 15 s, 55 °C for 30 s, and 68 °C for 30 s, with a final extension at 68 °C for 5 min. PCR products were resolved on a 1% agarose gel stained with SYBR Safe.

Colonies showing a correctly sized PCR band were cultured in serial dilutions in LB medium containing chloramphenicol to isolate clonal populations. Positive PCR amplicons were purified and submitted for Sanger sequencing to confirm precise integration of the StrepTag II without additional mutations. The resulting strains—WT,*pgk*::StrepTagII, Δ*dnaKJ*,*pgk*::StrepTagII, Δ*tig*,*pgk*::StrepTagII—were stored as glycerol stocks at −80 °C for future use.

### PGK expression and purification

Above-mentioned strains were grown at 30 °C overnight to generate saturated overnight cultures, and then 4 × 1 L of LB day culture were inoculated at 0.05 initial $OD_{600}$. Cells were grown at 30 °C with agitation (220 rpm) to a final OD 600 of 0.8, followed by centrifugation at 4000× g for 15 min at 4 °C. Supernatants were removed, and cell pellets were stored overnight at −80 °C until further use.

### Strep-Tactin XT 4Flow high-capacity affinity purification

Cell pellets were resuspended in 10 mL of Buffer W (100 mM Tris-HCl, pH 8.0, 150 mM NaCl, 1 mM EDTA) supplemented with addition DNase I (0.1 mg/ml), PMSF (500 µM), E-64 (15 µM), and Bestatin (50 µM), lysed by sonication (55% amplitude, 5 s on /5 s off pulses; total time adjusted based on culture volume), and clarified by centrifugation at 16,000×g for 15 min at 4 °C to remove cellular debris. The clarified lysate was subjected to affinity purification using Strep-Tactin XT 4Flow high-capacity gravity flow columns (IBA Lifesciences). Columns were first prepared by removing the top cap, followed by the bottom cap, allowing the storage buffer to drain without introducing air into the resin bed. Columns were equilibrated with 2 × 1 column volume (CV) of Buffer W (100 mM Tris-HCl, pH 8.0, 150 mM NaCl, 1 mM EDTA), freshly prepared and pre-chilled on ice. A column volume was defined based on the resin volume used (typically 300–500 µL for 4 L cultures). The clarified lysate was applied to the equilibrated column and incubated at 4 °C for 1 h to allow binding. The flow-through was collected for subsequent SDS-PAGE analysis. Columns were then washed with 5 × 1 CV of Buffer W, and each wash fraction was collected separately for analysis. Bound proteins were eluted using 6 × 0.5 CV of Buffer BXT (100 mM Tris-HCl, pH 8.0, 150 mM NaCl, 1 mM EDTA, 50 mM biotin), freshly diluted to 1× from a 10× stock stored at 4 °C. Elution fractions were analyzed by SDS-PAGE.

### Regeneration and storage of Strep-Tactin® XT 4Flow® columns

For regeneration, columns were washed with 15 CV of Buffer XT-R (3 M MgCl₂, IBA Lifesciences) at room temperature. Following regeneration, column operability was confirmed by applying 1× Buffer R (100 mM Tris-HCl pH 8.0, 150 mM NaCl, 1 mM EDTA, 1 mM HABA), which induces an orange color shift if regeneration was successful.

Immediately after confirmation, columns were washed with 8 CV of Buffer W. For storage, 2 mL of Buffer W was overlaid on the column, and both caps were replaced (top first, then bottom). Columns were stored at 2–8 °C.

### Anion exchange chromatography of PGK

Following affinity purification, phosphoglycerate kinase (PGK) was further purified using anion exchange chromatography on an ÄKTA™ Go FPLC system (Cytiva). The eluate from the Strep-Tactin XT 4Flow column was diluted 1:2 with Buffer A (20 mM Tris-HCl, pH 8.0, 10 mM NaCl) to reduce salt concentration prior to loading onto a HiTrap™ Q HP column (1 mL, Cytiva), pre-equilibrated with Buffer A.

Protein was loaded at a constant flow rate of 1 mL/min, and unbound material was washed off using 5 column volumes (CV) of Buffer A. Bound proteins were eluted using a linear gradient of 0–1 M NaCl over 20 CV using Buffer B (1 M NaCl). Elution was monitored by UV absorbance at 280 nm, and fractions were collected throughout the gradient.

Fractions corresponding to PGK, as determined by SDS-PAGE, were pooled and concentrated using a 10 kDa MWCO centrifugal filter unit (Amicon® Ultra, Millipore). The protein was exchanged into 20 mM Tris-HCl pH 8.0, 10 mM NaCl by repeated dilution-concentration steps, and then concentrated until [PGK] concentration was 5-10 μM, and then pure glycerol was added to a final concentration of 10% v/v to generate storage buffer (18 mM Tris-HCl pH 8.0, 9 mM NaCl, 10% v/v glycerol). Protein samples were then aliquoted, flash-frozen by immersion in liquid nitrogen, and stored at −80 °C until future use.

### In vitro translation and purification of PGK

Phosphoglycerate kinase (PGK) was synthesized using the PUREfrex® 2.1 in vitro translation kit (GeneFrontier Corporation) following the manufacturer's instructions. The template used was a plasmid encoding PGK cloned into a pET-24 vector with a C-terminal His$_6$-tag. The in vitro transcription-translation reaction was carried out at 30 °C for 3 h.

Following the reaction, the expressed PGK was purified using Ni-NTA magnetic beads (ThermoFisher) under native conditions. The eluate was dialyzed overnight at 4 °C into buffer containing 20 mM Tris-HCl (pH 8.0), 10 mM NaCl. Protein expression and purification were confirmed by SDS-PAGE and western blot using an anti-His primary antibody (ThermoFisher) and HRP-conjugated secondary antibody. Protein concentration was determined by measuring absorbance at 280 nm using a NanoDrop spectrophotometer.

### PGK activity assay for in vitro translated enzyme

PGK enzymatic activity was assessed using a coupled spectrophotometric assay that monitors the oxidation of NADH at 340 nm. The reaction couples PGK activity with glyceraldehyde-3-phosphate dehydrogenase (GAPDH), where PGK catalyzes the conversion of 3-phosphoglycerate (3-PG) and Mg-ATP to 1,3-bisphosphoglycerate, which is then converted to glyceraldehyde-3-phosphate by GAPDH with concomitant oxidation of NADH to NAD$^{\cdot}$.

The reaction mixture contained 100 mM Tris-HCl (pH 8.0), 2 mM MgCl$_2$, 0.2 mM NADH, 5 mM Mg-ATP, 0.4 μM GAPDH (Millipore Sigma, G2267), and 0.5 mM EDTA in a final volume of 200 μL per well in a 96-well plate. A final concentration of 5 mM 3-PG was used in all reactions. PGK was added to a final concentration of 0.00025 μM to initiate the reaction. PGK synthesized in vitro using the PUREfrex® 2.1 system and purified via Ni-NTA affinity purification was compared to PGK expressed in E. coli using the same plasmid construct and similarly purified.

Absorbance at 340 nm was recorded every 15 s for 30 min using a SpectraMax iD3 microplate reader (Molecular Devices). Control reactions lacking PGK were used to subtract background absorbance. Initial velocities were derived from the slope of the linear portion of the time series, and then divided by the slope of an NADH calibration curve (to convert from A$_{340}$/min to [NADH]/min). Unimolecular rate constants were then derived by dividing the maximal rate by enzyme concentration.

### Circular dichroism (CD) spectroscopy

Circular dichroism spectroscopy was performed to assess the secondary structure and thermal stability of PGK purified from wild-type (WT), Δ*dnaKJ*, and Δ*tig* E. coli strains. All CD measurements were carried out on an AVIV model 420 circular dichroism spectrometer equipped with a thermoelectric temperature control unit.

For far-UV CD scans, protein samples were buffer-exchanged into 10 mM sodium phosphate (pH 7.5) and diluted to a final concentration of 2 μM. Spectra were recorded from 250 to 195 nm at 25 °C in a 1 mm pathlength quartz cuvette. Each spectrum represents the average of ten scans. Data were acquired with a bandwidth of 1 nm, a step size of 0.5 nm, and an averaging time of 3 s per point.

Thermal denaturation experiments were conducted by monitoring ellipticity at 220 nm while increasing the temperature from 25 °C to 95 °C at a rate of 1 °C/min. Melting temperatures (Tm) were determined by fitting the resulting thermal unfolding curves to a sigmoidal Boltzmann function using GraphPad Prism software.

### PGK enzymatic activity assay for StrepTagged PGK from WT, Δ*dnaKJ*, and Δ*tig* backgrounds

The reaction mixture contained 100 mM Tris-HCl (pH 8.0), 2 mM MgCl$_2$, 0.2 mM NADH, 5 mM Mg-ATP, 0.4 μM GAPDH (Millipore Sigma, G2267), and 0.5 mM EDTA in a final volume of 200 μL per well in a 96-well plate. Final concentrations of 3-PG in individual wells were 0, 0.1, 0.25, 0.5, 1, 1.5, 2.5, and 5 mM, adjusted by varying the volumes of 50 mM 3-PG stock and assay buffer (100 mM Tris-HCl, 2 mM MgCl$_2$). Each well contained 100 μL of the master mix (without enzyme), to which 100 μL of purified PGK (final enzyme concentration, 0.02 μM) was added to initiate the reaction. GAPDH stock was prepared by dissolving 5.3 mg of GAPDH in 925 μL of assay buffer to generate a 40 μM stock, which was aliquoted and stored at −80 °C. Absorbance at 340 nm was monitored every 30 s for 30 min using a SpectraMax iD3 microplate reader (Molecular Devices). An NADH standard curve (0–100 nmol/well) was included on each plate for quantification. Background absorbance (no enzyme) was subtracted from each reading. Initial velocities were derived from the slope of the linear portion of the time series, and then divided by the slope of an NADH calibration curve (to convert from A$_{340}$/min to [NADH]/min). Michaelis–Menten kinetics were derived by plotting initial rates against substrate concentrations and fitting the data to the Michaelis–Menten equation using nonlinear regression analysis.

### Intact mass spectrometry analysis

Intact mass spectrometry was performed to assess the molecular weight and integrity of PGK purified from wild-type, Δ*dnaKJ*, and Δ*tig* E. coli backgrounds. Protein samples were buffer-exchanged into 100 mM ammonium acetate (pH ~7.0) using Amicon® Ultra-

0.5 mL centrifugal filters (10 kDa MWCO, Millipore) and adjusted to a final concentration of ~5 μM.

Analyses were carried out on a Waters ACQUITY UPLC system coupled to a Xevo G2-XS QTof mass spectrometer (Waters Corporation). Samples were injected onto a reverse-phase C4 column and separated under denaturing conditions using a linear gradient of acetonitrile in 0.1% formic acid. The mass spectrometer was operated in positive ion mode, and data were acquired across an $m/z$ range of 600–3000.

Raw spectra were deconvoluted using MassLynx and BioMassLynx deconvolution software to generate zero-charge intact masses. The observed molecular weights were compared with theoretical values based on the PGK sequence, including any affinity tags, to confirm protein identity.

### KatG activity assay

To assess catalase (KatG) activity, *E. coli* strains corresponding to wild-type (WT), Δ*dnaKJ*, and Δ*tig* backgrounds were cultured under identical conditions described above. Cells were harvested by centrifugation, and pellets were resuspended and lysed by sonication in ice-cold 50 mM potassium phosphate ($KP_i$, pH 7.0), which was supplemented with 1 mM PMSF and 0.1 mg/ml DNaseI. Lysates were clarified by centrifugation at $16,000 \times g$ for 15 min at 4 °C. Protein concentrations were determined using the Pierce™ BCA Protein Assay Kit (Thermo Scientific), and all samples were normalized to a final concentration of 1 mg/mL prior to assay.

KatG activity was measured using a luminescence-based spectrophotometric assay in 50 mM $KP_i$, pH 7.0. The assay monitored the breakdown of hydrogen peroxide ($H_2O_2$) by KatG, detected via chemiluminescence. Reactions were set up in triplicate in 96-well plates. Briefly, 10 μL of a 360 mM $H_2O_2$ stock solution (final concentration 20 mM; prepared from a 9.8 M $H_2O_2$ stock, Sigma-Aldrich) was added to 170 μL of the lysates, which were diluted to final protein concentrations of 0.2, 0.3, or 0.4 mg/mL with 50 mM $KP_i$, pH 7.0. After a 1-min incubation, 10 μL of luminol (final concentration 0.1 mM; prepared from an 11.3 mM stock solution in the SuperSignal™ West Femto Maximum Sensitivity Substrate Kit, Thermo Scientific) was added. A 1:100 dilution of Strep-HRP conjugate (iba lifesciences) was prepared fresh, and 10 μL was added to each reaction immediately following luminol addition.

Plates were immediately read for chemiluminescence decay using a SpectraMax iD3 microplate reader (Molecular Devices), with luminescence recorded over time across all wavelengths. Appropriate controls—lacking $H_2O_2$ or lysate—were included in every run to account for background signal. Relative KatG activities were quantified by normalizing luminescence signals to those obtained from the WT control.

### Proteomics analysis of KatG abundance

To assess KatG protein abundance, a proteomics workflow was performed using the same normalized lysates prepared for the activity assay. Equal amounts of total protein were subjected to reduction, alkylation, and trypsin digestion following a standard in-solution digest sample preparation protocol. Peptides were desalted using C18 solid-phase extraction and dried under vacuum prior to LC-MS/MS analysis. Peptide mixtures were analyzed using an Orbitrap Ascend Tribrid mass spectrometer coupled to a nanoLC system. Samples were separated on a reversed-phase nanoLC column with a linear acetonitrile gradient in 0.1% formic acid. MS data were acquired in data-dependent acquisition (DDA) mode. Raw MS data were processed using FragPipe v22.0, incorporating MSFragger for database searching and Philosopher for validation. Label-free quantification (LFQ) was performed using MaxLFQ intensities, and statistical comparisons between groups (WT, Δ*dnaKJ*, and Δ*tig*) were conducted to calculate fold changes and associated *P* values.

## Data availability

The datasets and computer code produced in this study are available in the following databases: Mass spectrometry proteomics data: PRIDE PXD064568. Cut-site quantifications: Zenodo 16879364.

The source data of this paper are collected in the following database record: biostudies:S-SCDT-10_1038-S44320-025-00166-6.

## Peer review information

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

## Acknowledgements

The authors would like to acknowledge Edgar Manriquez-Sandoval for assistance with using the FLiPPR software package, Yingzi Xia for providing purified His-Tagged PGK protein expressed in *E. coli*, Phil Mortimer for supporting the mass spectrometry core facility at Johns Hopkins Chemistry, and Katherine Tripp for training and support with circular dichroism experiments. The authors thank Ulrich Hartl for kindly providing the *E. coli* strains used in this study, and Bernd Bukau for comments and feedback on the manuscript. SDF acknowledges support from the NIH Director's New Innovator Award (DP2-GM140926), from the National Science Foundation (MCB-2045844), from a Camille Dreyfus Teacher-Scholar Award, and from a Cottrell Scholars Award (Research Corporation for Science Advancement). MF was an Amgen Scholar and acknowledges support from Amgen.

## Author contributions

**Divya Yadav**: Conceptualization; Data curation; Formal analysis; Validation; Investigation; Visualization; Methodology; Writing—original draft; Writing—review and editing. **İdil I Demiralp**: Investigation; Writing—review and editing. **Mark Fakler**: Investigation; Writing—review and editing. **Stephen D Fried**: Conceptualization; Formal analysis; Supervision; Funding acquisition; Investigation; Visualization; Writing—original draft; Project administration; Writing—review and editing.

Source data underlying figure panels in this paper may have individual authorship assigned. Where available, figure panel/source data authorship is listed in the following database record: biostudies:S-SCDT-10_1038-S44320-025-00166-6.

## Disclosure and competing interests statement

The authors declare no competing interests.

# Expanded View Figures

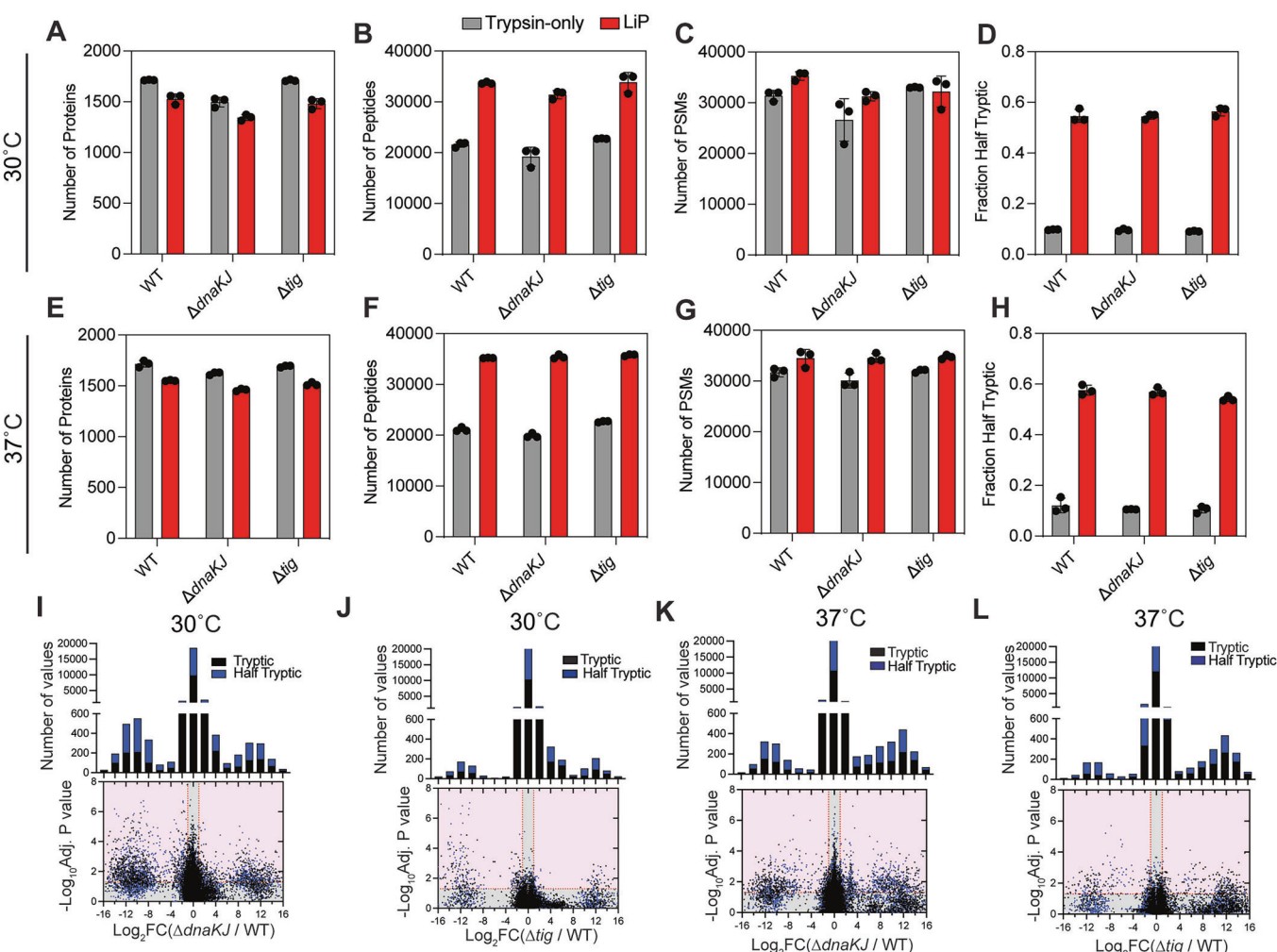

**Figure EV1. Quality control metrics and peptide type distributions in LiP-MS analysis across chaperone knockout strains.**

Number of proteins (A, E), peptides (B, F), and peptide-spectrum matches (PSMs) (C, G) identified per sample injection/bio-replicate from WT, ΔdnaKJ, and Δtig strains at 30 °C (top row) and 37 °C (bottom row) ($n = 3$). LiP samples (red bars) were subjected to limited proteolysis using Proteinase K followed by trypsin digestion, whereas trypsin-only controls (gray bars) underwent only tryptic digestion. Each dot represents a biological replicate ($n = 3$). Bars represent means and error bars represent standard deviation. (D, H) Fraction of peptides that are half-tryptic (i.e., one non-tryptic terminus attributed to Proteinase K cleavage) in each sample. (I–L) Volcano plots showing changes in cut-sites abundance (half-tryptic in blue; tryptic in black) between WT and chaperone knockout strains: (I) ΔdnaKJ/WT at 30 °C, (J) Δtig/WT at 30 °C, (K) ΔdnaKJ/WT at 37 °C, (L) Δtig/WT at 37 °C. Peptides with >2-fold change and $P < 0.05$ (Welch's $t$ test following protein-wise Benjamini–Hochberg FDR correction) are highlighted in pink-shaded regions. Insets above each volcano plot show the distribution of tryptic and half-tryptic peptides across $\log_2$ fold-change bins. All peptide quantifications are available in the associated Zenodo deposition (10.5281/zenodo.16879364).

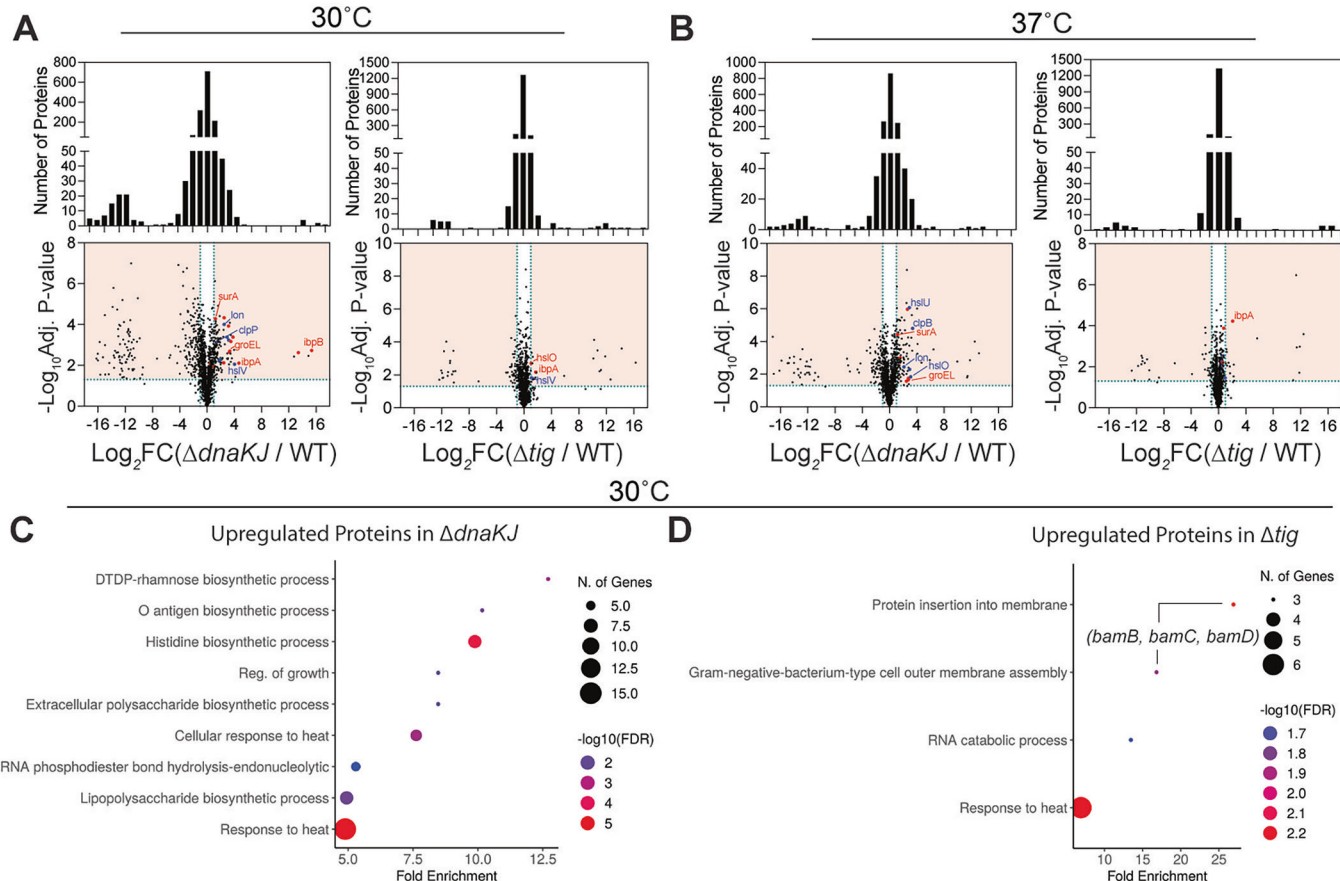

**Figure EV2. Global Protein abundance changes and GO enrichment analyses in Δ*dnaKJ* and Δ*tig* strains at 30 °C and 37 °C.**

(**A, B**) Volcano plots showing differential protein abundance in Δ*dnaKJ* (left panels) and Δ*tig* (right panels) relative to WT at 30 °C (**A**) and 37 °C (**B**). Each dot represents an individual protein. Significantly altered proteins (log$_2$ fold change > 1 or < −1; adjusted $P$ < 0.05; as calculated from Fragpipe) are highlighted in the shaded regions. Histograms above each plot display the distribution of fold changes. Molecular chaperones are labeled in red, proteases in blue (see Source Data for Fig. EV2A,B). (**C, D**) Gene Ontology (GO) Biological Process enrichment for proteins significantly upregulated in Δ*dnaKJ* (**C**) and Δ*tig* (**D**) at 30 °C. Dot size represents the number of genes enriched in each term; color indicates FDR-adjusted significance, as calculated by ShinyGo v.0.80. Δ*dnaKJ* cells show enriched processes related to heat response, biosynthesis, and polysaccharide metabolism, while Δ*tig* upregulated proteins are enriched in membrane insertion, cell envelope assembly, and RNA catabolism. Source data are available online for this figure.

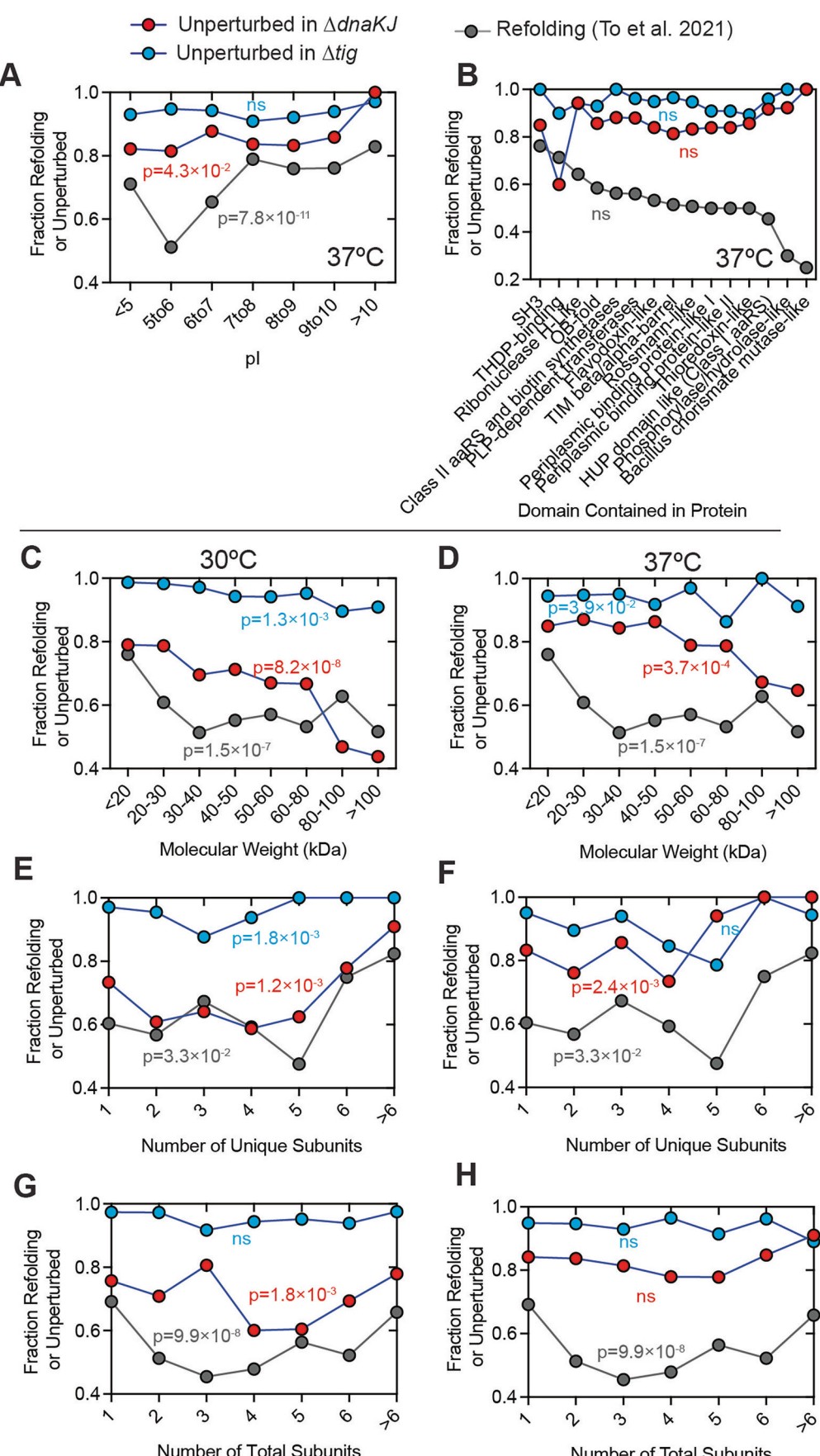

**Figure EV3.   Comparison of chaperone dependency during primary folding and in vitro refolding.**

Fraction of proteins that refold in vitro following chemical denaturation (gray; data from To et al, 2021) versus those that remain structurally unperturbed in vivo in either Δ*dnaKJ* (red) or Δ*tig* (blue) backgrounds, measured using LiP-MS at 30 °C (left panels) and 37 °C (right panels). Proteins are grouped by various properties: (**C**, **D**) Molecular weight (kDa); (**E**, **F**) Number of unique subunits; (**G**, **H**) Total number of subunits; (**A**) Isoelectric point (pI) at 37 °C; (**B**) Fold-type domain annotations according to ECOD X-group, at 37 °C. *P* values calculated with the chi-square test (see Source Data for Fig. EV3). Source data are available online for this figure.

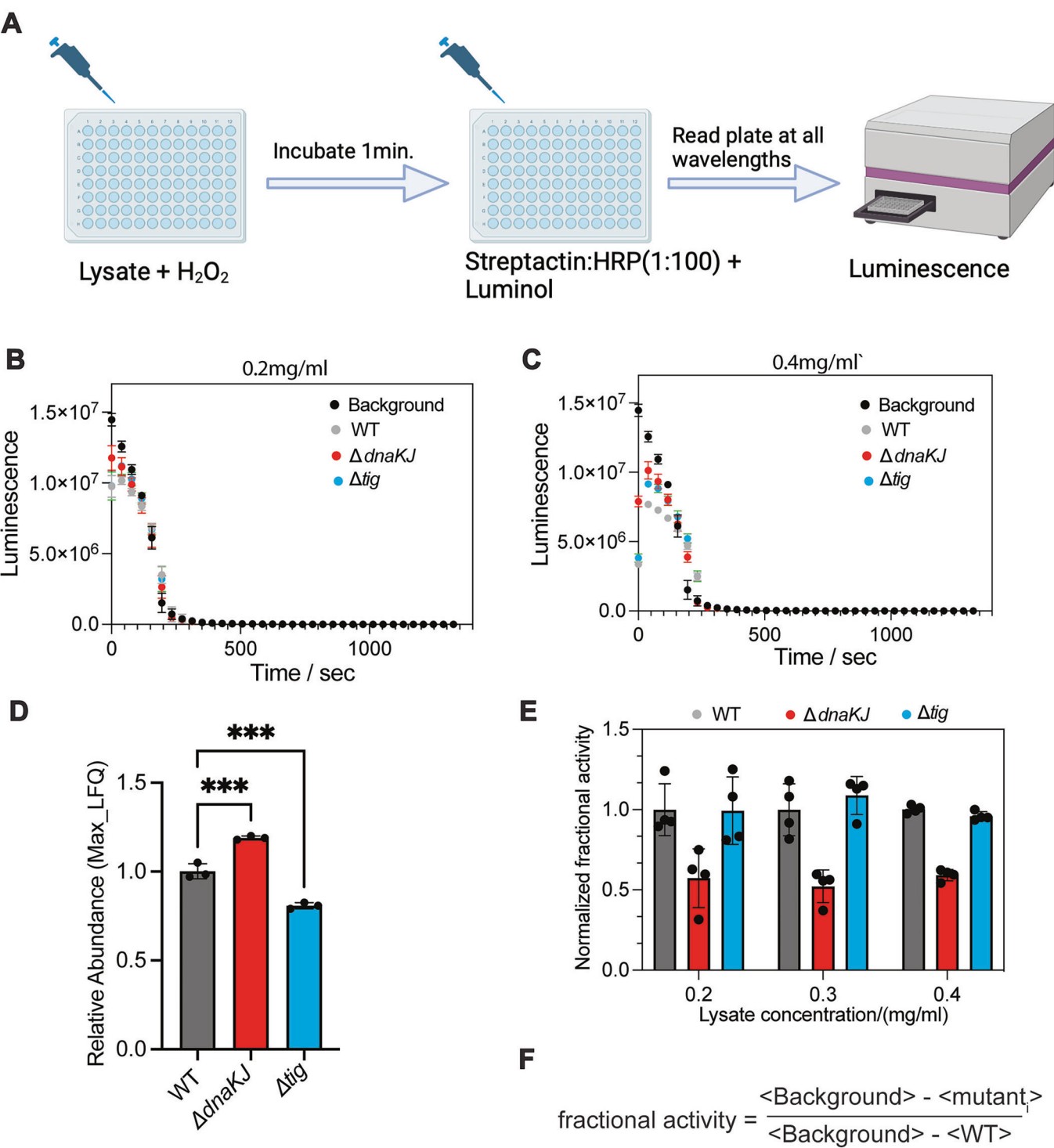

◀   **Figure EV4.   Luminescence-based catalase activity assay confirms functional impairment of KatG in Δ*dnaKJ* background.**

(A) Schematic of the assay workflow. Clarified lysates were incubated with $H_2O_2$ for 1 min, after which residual $H_2O_2$ was detected by adding streptavidin-HRP and luminol, with luminescence measured in real-time. (B, C) Raw luminescence decay curves over time for KatG activity in WT, Δ*dnaKJ*, and Δ*tig* lysates at two concentrations for 4 biological replicates (n = 4): 0.2 mg/mL (B) and 0.4 mg/mL (C). Background control lacking lysate is shown in black. Points represent means and error bars represent standard deviation (see Source Data for Fig. EV4B,C). (D) Relative abundance of KatG across strains as measured by MaxLFQ protein quantification (n = 3). P values were calculated using ordinary one-way ANOVA followed by Tukey's multiple comparisons test (α = 0.05). Asterisks indicate statistical significance as follows: ***$P < 0.001$ (see Source Data for Fig. EV4D). (E) Normalized fractional catalase activity at varying lysate concentrations. Activity in Δ*dnaKJ* is consistently reduced across all concentrations, whereas Δ*tig* retains near-WT activity. Bars represent means and error bars represent standard deviation (see Source Data for Fig. EV4E). (F) Formula used to calculate fractional activity relative to WT and background controls. Source data are available online for this figure.

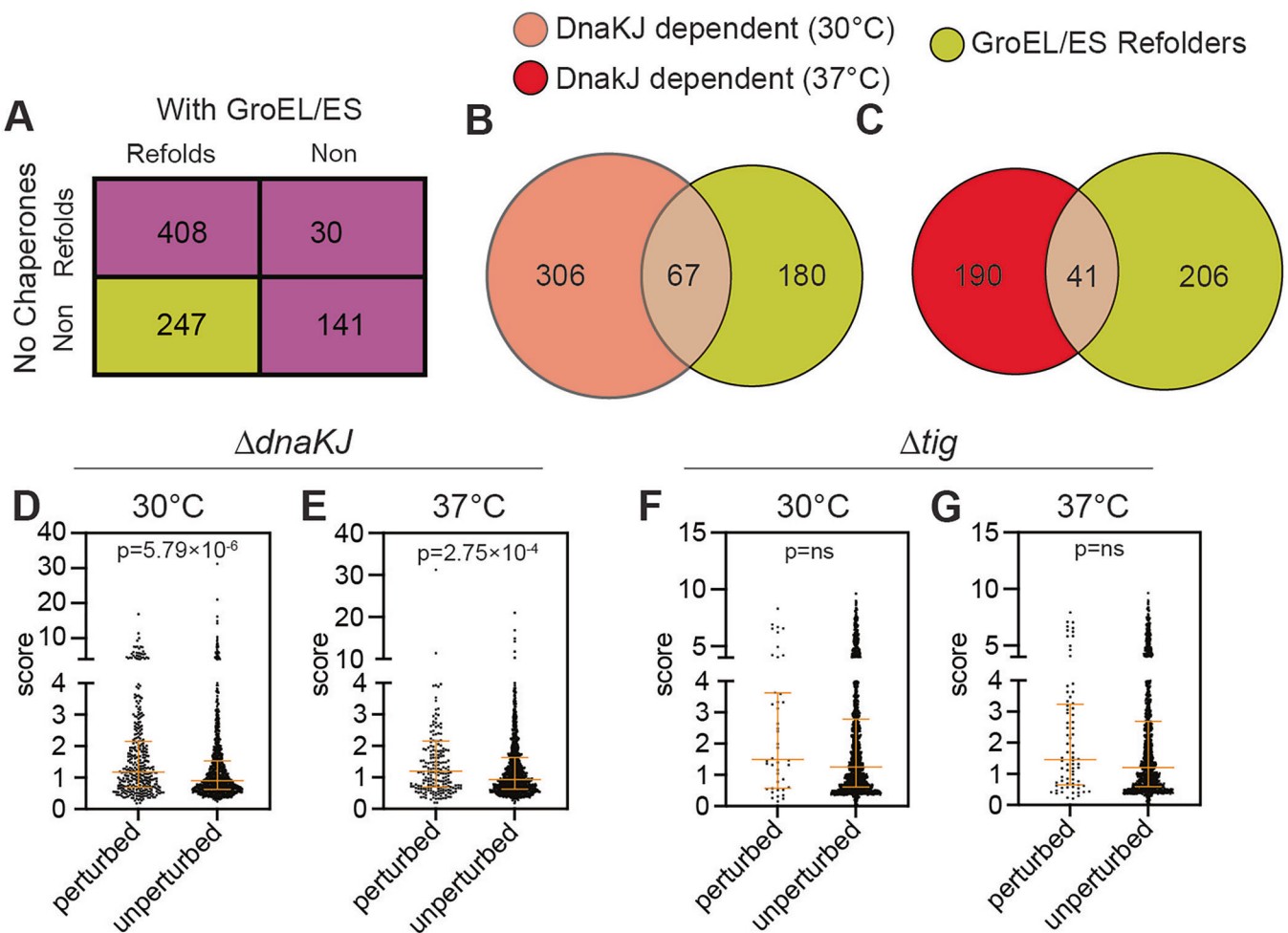

**Figure EV5. Comparison of GroEL/ES-assisted refolding with DnaKJ dependence at different temperatures and comparison of in vivo Δ*dnaKJ* perturbation data with chaperone binding scores from Galmozzi et al, 2025.**

(A) Contingency table from chaperone refolding experiments (To et al, 2022) that defines which proteins are categorized as GroEL-dependent refolders (see Source Data for Fig. EV5A). (B) Overlap between in vitro GroEL-dependent refolders ($n = 247$) and in vivo DnaKJ-dependent substrates ($n = 373$) identified by structural perturbation in Δ*dnaKJ* cells at 30 °C. (C) Same comparison as in (B), but using in vivo Δ*dnaKJ* LiP-MS data at 37 °C ($n = 231$) (see Source Data for Fig. EV5B,C). (D–G) Scatter plots show chaperone binding scores (y-axis), reflecting binding affinity to the DnaKJ chaperone from Galmozzi et al, 2025, for proteins classified as perturbed ("p") or unperturbed ("u") in our in vivo LiP-MS dataset. (D) Δ*dnaKJ* strain at 30 °C ($n_p = 373$, $n_u = 974$). (E) Δ*dnaKJ* strain at 37 °C ($n_p = 231$, $n_u = 1120$). (F) Δ*tig* strains at 30 °C ($n_p = 45$, $n_u = 1298$). (G) Δ*tig* strains at 37 °C ($n_p = 81$, $n_u = 1271$). *P* values (shown) were calculated using the non-parametric Kolmogorov–Smirnov test. Horizontal orange lines represent median ± interquartile range (see Source Data for Fig. EV5D,E and EV5F,G). Source data are available online for this figure.

