## [Peer Review File · Molecular Systems Biology]

Chaperone Dependency during Biogenesis Does Not Correlate with Chaperone Dependency during Refolding

Divya Yadav, İdil Demiralp, Mark Fakler, and Stephen Fried

Corresponding author(s): Stephen Fried (sdfried@jhu.edu)

Review Timeline:

Submission Date:	24th Jun 25
Editorial Decision:	23rd Jul 25
Revision Received:	19th Aug 25
Editorial Decision:	15th Sep 25
Revision Received:	19th Sep 25
Accepted:	26th Sep 25

Editor: Poonam Bheda

Transaction Report:

23rd Jul 2025

Manuscript Number: MSB-2025-13201

Title: Chaperone Dependency during Biogenesis Does Not Correlate with Chaperone Dependency during Refolding

Dear Prof Fried,

Thank you for the submission of your manuscript to Molecular Systems Biology. We have now received feedback from the three reviewers who agreed to evaluate your manuscript. As you will see from the reports below, the referees acknowledge the interest of the study and are overall supporting publication of your work pending appropriate revisions.

I think that the recommendations of the reviewers are rather clear and I therefore do not see the need to repeat the comments listed below. All issues raised would need to be satisfactorily addressed. Please let me know in case you would like to discuss in further detail any of the any of the reviewer comments or your proposed revisions, I would be happy to schedule a call.

We require:

- 1) A .docx formatted version of the manuscript text (including legends for main figures, EV figures and tables). Please make sure that the changes are highlighted to be clearly visible. Alternatively you may choose to submit your manuscript as a LaTeX file.
- 2) Individual production quality figure files as .eps, .tif, .jpg (one file per figure). For guidance, download the 'Figure Guide PDF' (<https://www.embopress.org/page/journal/17574684/authorguide#figureformat>).
- 3) At EMBO Press we ask authors to provide source data for the main figures. Our source data coordinator will contact you to discuss which figure panels we would need source data for and will also provide you with helpful tips on how to upload and organize the files.
- 4) A .docx formatted letter INCLUDING the reviewers' reports and your detailed point-by-point responses to their comments. As part of the EMBO Press transparent editorial process, the point-by-point response is part of the Peer Review File (PRF), which will be published alongside your paper.
- 5) A complete author checklist, which you can download from our author guidelines (<https://www.embopress.org/page/journal/17574684/authorguide#submissionofrevisions>). Please insert information in the checklist that is also reflected in the manuscript. The completed author checklist will also be part of the PRF.
- 6) Please note that all corresponding authors are required to supply an ORCID ID for their name upon submission of a revised manuscript.
- 7) It is mandatory to include a 'Data Availability' section after the Materials and Methods. Before submitting your revision, primary datasets produced in this study need to be deposited in an appropriate public database, and the accession numbers and database listed under 'Data Availability'. Please remember to provide a reviewer password if the datasets are not yet public (see <https://www.embopress.org/page/journal/17574684/authorguide#dataavailability>).

In case you have no data that requires deposition in a public database, please state so in this section as follows: "This study includes no data deposited in external repositories". Note that the Data Availability Section is restricted to new primary data that are part of this study.

- 8) All Materials and Methods need to be described in the main text using our 'Structured Methods' format, which is required for all research articles. According to this format, the Methods section includes a Reagents and Tools Table (listing key reagents, experimental models, software and relevant equipment and including their sources and relevant identifiers) followed by a Methods and Protocols section describing the methods using a step-by-step protocol format. The aim is to facilitate adoption of the methodologies across labs. Please upload the Reagents and Tools table as a separate document when submitting your revised manuscript. More information on how to adhere to this format as well as a downloadable template (.docx) for the Reagents and Tools Table can be found in our author guidelines:
<https://www.embopress.org/page/journal/17444292/authorguide#structuredmethods>

9) For data quantification: please specify the name of the statistical test used to generate error bars and p-values, the number (n) of independent experiments (specify technical or biological replicates) underlying each data point and the test used to calculate p-values in each figure legend. The figure legends should contain a basic description of n, p-values and the test applied. Graphs must include a description of the bars and the error bars (s.d., s.e.m.). Please provide exact p-values (in either the figure or figure legend).

10) Our journal encourages inclusion of *data citations in the reference list* to directly cite datasets that were re-used and obtained from public databases. Data citations in the article text are distinct from normal bibliographical citations and should directly link to the database records from which the data can be accessed. In the main text, data citations are formatted as follows: "Data ref: Smith et al, 2001" or "Data ref: NCBI Sequence Read Archive PRJNA342805, 2017". In the Reference list, data citations must be labeled with "[DATASET]". A data reference must provide the database name, accession number/identifiers and a resolvable link to the landing page from which the data can be accessed at the end of the reference. Further instructions are available at .

11) We replaced Supplementary Information with Expanded View (EV) Figures and Tables that are collapsible/expandable online. EV Figures should be cited as 'Figure EV1, Figure EV2' etc... in the text and their respective legends should be included in the main text after the legends of regular figures.

- Additional Tables/Datasets should be labeled and referred to as Table EV1, Dataset EV1, etc. Legends should be provided in a separate tab in case of .xls files. Alternatively, the legend can be supplied as a separate text file (README) and zipped together with the Table/Dataset file.

<https://www.embopress.org/page/journal/17574684/authorguide#expandedview>

12) Author contributions: CRediT has replaced the traditional author contributions section because it offers a systematic machine-readable author contributions format that allows for more effective research assessment. Please remove the Authors Contributions from the manuscript and use the free text boxes beneath each contributing author's name in our system to add specific details on the author's contribution. More information is available in our guide to authors.

13) Disclosure statement and competing interests: We updated our journal's competing interests policy in January 2022 and request authors to consider both actual and perceived competing interests. Please review the policy <https://www.embopress.org/competing-interests> and update your competing interests if necessary.

14) Every published paper now includes a 'Synopsis' to further enhance discoverability. Synopses are displayed on the journal webpage and are freely accessible to all readers. They include a short stand first (maximum of 300 characters, including space) as well as 2-5 one-sentences bullet points that summarizes the paper. Please write the bullet points to summarize the key NEW findings. They should be designed to be complementary to the abstract - i.e. not repeat the same text. We encourage inclusion of key acronyms and quantitative information (maximum of 30 words / bullet point). Please use the passive voice. Please attach these in a separate file or send them by email, we will incorporate them accordingly.

Please note that these would be the final versions and changes during proofing are usually not allowed.

15) As part of the EMBO Publications transparent editorial process initiative (see our policy here:

https://www.embopress.org/transparent-process#Review_Process), Molecular Systems Biology will publish online a Peer Review File (PRF) to accompany accepted manuscripts.

In the event of acceptance, this file will be published in conjunction with your paper and will include the anonymous referee reports, your point-by-point response and all pertinent correspondence relating to the manuscript. Let us know whether you agree with the publication of the PRF and as here, if you want to remove or not any figures from it prior to publication.

Please note that the Author checklist will be published at the end of the PRF.

Molecular Systems Biology has a "scooping protection" policy, whereby similar findings that are published by others during review or revision are not a criterion for rejection. Should you decide to submit a revised version, I do ask that you get in touch after three months if you have not completed it, to update us on the status.

Yours sincerely,

Poonam Bheda, PhD
Scientific Editor

Reviewer #1:

In this study, the authors ask a deceptively simple question: does the set of *E. coli* proteins that rely on cytosolic chaperones during their primary biogenesis *in vivo* match the set that need those same chaperones to refold from a fully-denatured state *in vitro*? To answer it they combine (i) quantitative limited-proteolysis mass-spectrometry (LiP-MS) on whole-cell lysates from strains lacking Trigger Factor (Tig) or Hsp70/Hsp40 (DnaKJ) with (ii) a rigorous re-analysis of their own 2022 proteome-wide refolding dataset. Selected enzymes (phosphoglycerate kinase, catalase) are purified to biochemically validate the structural proteomics calls.

Key Conclusions:

- 1) DnaKJ is required for native conformations of ~28 % of the soluble proteome at 30 {degree sign}C but only ~17 % at 37 {degree sign}C, whereas Tig deletion affects {less than or equal to}6 % of proteins at either temperature.
- 2) Proteins that depend on DnaKJ *in vivo* show little overlap (13-17 %) with proteins that require DnaKJ to refold *in vitro*, indicating distinct client sets for co-translational vs post-translational chaperone action.
- 3) Canonical "chaperone-non-refolders" such as PGK fold perfectly *in vivo* without Tig or DnaKJ, underscoring the power of vectorial, ribosome-guided folding.
- 4) DnaKJ dependence at low temperature is enriched for multi-domain proteins and holoproteins bearing covalently bound cofactors (heme, iron-sulfur, PLP). The authors propose that ATP-driven unfolding by DnaKJ helps these proteins escape kinetic traps that become self-resolvable at 37 {degree sign}C.
- 5) LiP-MS perturbations correlate with function: catalase (KatG) shows active-site structural changes in Δ dnaKJ lysates and a 35-40 % drop in specific activity.

Overall, I really enjoyed reading this paper. This group continues to be at the forefront of MS-based proteostasis research and I appreciate the huge effort that went into this work. Andy Truman

Minor comments:

- 1) GroEL is essential but highly induced in Δ dnaKJ. Could some DnaKJ clients be rerouted to GroEL? The authors should discuss this possibility.
- 2) Fig. 2E-F: please consider using colour-blind-safe palettes as the red/green is indistinguishable.
- 3) The abstract exceeds 250 words-I would recommend trimming this down a bit.
- 4) Check for consistent italics for gene names (dnaK, tig) and Roman for proteins (DnaK).
- 5) Methods: please change "Eppendor Eporator" to "Eppendorf Eporator"

Reviewer #2:

In this work, the different roles of the DnaK and Trigger Factor (Tig) chaperones at helping the folding of newly synthesized proteins is analyzed *in vivo*, and compared with their roles in *in vitro* refolding experiments. Overall, the manuscript is well-written, and it conveys a message that, although in my opinion less definitive than what the authors claim, represent a clear step forward and a somewhat provocative result.

Before I lay out my criticisms, I want to be clear on the fact that I am not asking for new experiments at this stage. My own rule is that new experiments should be performed only if necessary controls are lacking. In this case, the authors will be able to address my comments by casting their conclusions in a possibly milder way. Of course, should they be willing to perform the consequent experiments, they are most welcome, but I do not deem that necessary.

Here come my major comments, in no particular order:

1) I overall agree that the difference between 30{degree sign}C and 37{degree sign}C might be due to the increased "rigidity" of misfolded proteins at lower temperatures. At the same time, the authors mention these as "non-stress" temperatures. This should be given better context. Indeed, 37{degree sign}C is physiological in the gut, but is 30{degree sign}C physiological outside of it? Is *E. coli* adapted (non-stress) over that window, or does it need some time to adapt as its environment changes (e.g. is ingested in or is expelled from the gut)? Likewise, other animals might have lower or higher physiological temperatures (e.g. birds are physiological at 39{degree sign}C, which can normally be used as a stressor in cultured *E. coli*; reptiles have likely a lesser body temperature). In this respect, the authors could use a word of caution when comparing results at the two temperatures to convey a message about the physiological roles.

2) The authors have compared the DeltaDnaK and the DeltaTig with the WT, and looked for overlaps (with respect to the >1 cuts). Yet, since their role is not necessarily additive, an analysis of their direct comparison would have been also pretty telling. Any reasons not to do so?

3) The comparison with whole-proteome refolding (To et al, 2022) is very interesting, but what definitive statement can be made from it is less clear to me. In those experiments, the full proteome (lysate) was first denatured and then denaturant diluted while adding either GroELS or KJE. While this experiment tells about the intrinsic role of these chaperones, and as such is interesting of course, it meddles the waters a bit. Chaperones do not function, for the most part, as independent agents. KJE might hand-over its proteolysis to GroELS, or might be helped, in some cases such as sturdy aggregates, to ClpB (Hsp100). HtpG (Hsp90) might be relevant for a subset of substrates, such as kinases, refolding, while it might act independently or downstream of KJE. What about small heat shock proteins?

While in the in vitro (lysate) experiments all of them were most likely denatured and absent, so that only the added chaperones were relevant, in the in vivo Deltas they are present in background, and I am not completely sure their effects can be easily disentangled.

4) What said at point 3 applies even more for "non-refoldable" substrates. They were deemed non-refoldable when the denatured lysate was supplemented with either KJE or with GroELS. What if they needed the full chaperone network to be refoldable, including possibly being captured by sHSPs along the way?

5) I take some issue with the physiological relevance of refolding experiments, if the goal is to infer the role of chaperones. Let me be clear: To et al. 2022 is an excellent paper. But in the context of the present one, I am not so sure that it is the good way to argue about post-translational refolding (but to be fair: this is a criticism for 95% of the experiments in the literature). This is my beef with it:

I doubt that translated and folded proteins do suddenly unfold, unchecked, and then chaperones have to cope with them. Most likely, chaperones act immediately at the first signs of unfolding/misfolding, and maintain proteins native possibly even in mildly non-native conditions, in particular in the non-stressful region that the authors have investigated here.

I would like to authors to at least comment about this.

6) Last but not least: How can the authors clearly disentangle the misfolding of newly synthesized proteins from the one of proteins that are already present? What I mean is that should KJ also act as suggested in point 5, there could be some native state maintenance in the WT that is lacking in the DeltaKJ, so that there might be some mixture of co-translational and post-translational misfolding. Experiments to test this could be difficult but not impossible (like starting feeding isotope-modified cultures at some point, to distinguish newly synthesized proteins from the pre-existing ones). I am not asking for them, but maybe some discussions would be useful.

Minor:

here and there some passages could be smoother. Please re-read carefully with the eyes of the audience, not of the authors.

Fig.6E at page 24 should be 5E, if I am not mistaken.

Reviewer #3:

The protein folding problem is one of the fundamental questions in biology. Conventional folding studies, originating from Anfinsen's classical experiments, do not always reflect folding in vivo, since nascent polypeptides are synthesized vectorially from the N-terminus. This manuscript clearly demonstrates the difference between in vivo and in vitro folding, using a large-scale analysis with limited proteolysis mass spectrometry (LiP-MS). The study presented here is exactly what I hoped to see following the author's series of works using LiP-MS analyses. The results are highly interesting, and I have a few comments to clarify unclear points or to further strengthen the current conclusions.

1. If the group of structurally perturbed proteins in vivo, shown in this manuscript, is indeed cotranslationally assisted in folding by chaperones, especially DnaK, then these chaperones would bind to the ribosome during translation via the nascent chain. In that case, one could expect a correlation between the protein groups in this study and the proteome-wide data obtained through selective ribosome profiling, which would allow for a comprehensive analysis of the binding of TF and DnaK to nascent chains

(Galmozzi et al., 2025, cited in this manuscript). I encourage the authors to analyze this to strengthen the conclusions of this study.

2. The finding that structurally perturbed proteins in the Δ dnaKJ strain are more prominent at lower temperature (30 C) is indeed surprising.

2-a. Is there any difference in the perturbed proteins between 30 C and 37 C in wild-type E. coli?

2-b. Proteins that aggregate or are degraded at 37 C in the dnaKJ-KO strain may be more abundant than under 30 C conditions. Are there any effects of these?

Minor points:

3. p. 24: Fig. 6E should be Fig. 5E.

4. p. 25: CsdA should be CspA.

Dear Dr Bheda –

Please find attached our resubmitted manuscript. We are also providing some new files not present in our original submission: a Synopsis file and a Source Data directory that provides comprehensive reference data as well as the information needed to construct each of the figures. Below, we respond (in red) to all editorial remarks as well as to comments by the three reviewers.

We thank the editor and reviewers for their careful consideration, thoughtful feedback, and positive assessment.

With best wishes,

Stephen and Divya

23rd Jul 2025

Manuscript Number: MSB-2025-13201

Title: Chaperone Dependency during Biogenesis Does Not Correlate with Chaperone Dependency during Refolding

Dear Prof Fried,

Thank you for the submission of your manuscript to Molecular Systems Biology. We have now received feedback from the three reviewers who agreed to evaluate your manuscript. As you will see from the reports below, the referees acknowledge the interest of the study and are overall supporting publication of your work pending appropriate revisions.

I think that the recommendations of the reviewers are rather clear and I therefore do not see the need to repeat the comments listed below. All issues raised would need to be satisfactorily addressed. Please let me know in case you would like to discuss in further detail any of the any of the reviewer comments or your proposed revisions, I would be happy to schedule a call.

Thank you for these helpful reviews and pointers to complete the manuscript for publication. Please see our responses to the editorial comments below, in red.

We require:

1) A .docx formatted version of the manuscript text (including legends for main figures, EV figures and tables). Please make sure that the changes are highlighted to be clearly visible. Alternatively you may choose to submit your manuscript as a LaTeX file.

We supply both an unmarked .docx file and a traced .pdf file highlighting the differences from the initially submitted version. As instructed, all figures are now removed from the manuscript itself, and provided as separate high-res jpg files. The 6 figure legends and 5 EV figure legends are provided at the end of the manuscript file.

Thanks, we have done this.

Thanks, we have followed this. In the effort for full transparency, we have provided source data for every figure panel that shows data.

4) A .docx formatted letter INCLUDING the reviewers' reports and your detailed point-by-point responses to their comments. As part of the EMBO Press transparent editorial process, the point-by-point response is part of the Peer Review File (PRF), which will be published alongside your paper.

Thanks, that is included in this document, below.

5) A complete author checklist, which you can download from our author guidelines (<https://www.embopress.org/page/journal/17574684/authorguide#submissionofrevisions>). Please insert information in the checklist that is also reflected in the manuscript. The completed author checklist will also be part of the PRF.

Thanks, this is completed.

6) Please note that all corresponding authors are required to supply an ORCID ID for their name upon submission of a revised manuscript.

Thanks, we will provide ORCID for all authors.

7) It is mandatory to include a 'Data Availability' section after the Materials and Methods. Before submitting your revision, primary datasets produced in this study need to be deposited in an appropriate public database, and the accession numbers and database listed under 'Data Availability'. Please remember to provide a reviewer password if the datasets are not yet public (see <https://www.embopress.org/page/journal/17574684/authorguide#dataavailability>).

In case you have no data that requires deposition in a public database, please state so in this section as follows: "This study includes no data deposited in external repositories". Note that the Data Availability Section is restricted to new primary data that are part of this study.

Thanks, we have included such a section, it is titled "Data and materials availability." It reads as:

The mass spectrometry proteomics data have been deposited to the ProteomeXchange Consortium via the PRIDE partner repository with the dataset identifier PXD064568. Cut-site quantifications are available on Zenodo at /10.5281/zenodo.16879364. All reference data are data required to construct plots and figures are available as Source Data, associated with this paper. Strains used in this study (MC4100 wild-type, MC4100 Δ *tig*, MC4100 Δ *dnaKJ*) are available from a number of sources, but can also be supplied by the corresponding author upon request.

8) All Materials and Methods need to be described in the main text using our 'Structured Methods' format, which is required for all research articles. According to this format, the Methods section includes a Reagents and Tools Table (listing key reagents, experimental models, software and relevant equipment and including their sources and relevant identifiers) followed by a Methods and Protocols section describing the methods using a step-by-step protocol format. The aim is to facilitate adoption of the methodologies across labs. Please upload the Reagents and Tools table as a separate document when submitting your revised manuscript. More information on how to adhere to this format as well as a downloadable template (.docx) for the Reagents and Tools Table can be found in our author guidelines: <https://www.embopress.org/page/journal/17444292/authorguide#structuredmethods>

An example of a Method paper with Structured Methods can be found here: <https://www.embopress.org/doi/10.15252/msb.20178071>.

We have included subheading for "Methods Protocol" under the Materials and Methods section in the manuscript, and the Reagents and Tool Table is provided as a separate file.

9) For data quantification: please specify the name of the statistical test used to generate error bars and p-values, the number (n) of independent experiments (specify technical or biological replicates) underlying each data point and the test used to calculate p-values in each figure legend. The figure legends should contain a basic description of n, p-values and the test applied. Graphs must include a description of the bars and the error bars (s.d., s.e.m.). Please provide exact p-values (in either the figure or figure legend).

Thanks, we have taken care to make sure that this has been done, though we will appreciate an editorial check at the next round of review if any are missing.

10) Our journal encourages inclusion of *data citations in the reference list* to directly cite datasets that were re-used and obtained from public databases. Data citations in the article text are distinct from normal bibliographical citations and should directly link to the database records from which the data can be accessed. In the main text, data citations are formatted as follows: "Data ref: Smith et al, 2001" or "Data ref: NCBI Sequence Read Archive PRJNA342805, 2017". In the Reference list, data citations must be labeled with "[DATASET]". A data reference must provide the database name, accession number/identifiers and a resolvable link to the landing page from which the data can be accessed at the end of the reference. Further instructions are available at <https://www.embopress.org/page/journal/17574684/authorguide#referencesformat>.

Thanks for raising this important point. We have added three references to the reference list that are now referred to as Data refs.

11) We replaced Supplementary Information with Expanded View (EV) Figures and Tables that are collapsible/expandable online. EV Figures should be cited as 'Figure EV1, Figure EV2' etc... in the text and their respective legends should be included in the main text after the legends of regular figures.

- Additional Tables/Datasets should be labeled and referred to as Table EV1, Dataset EV1, etc. Legends should be provided in a separate tab in case of .xls files. Alternatively, the legend can be supplied as a separate text file (README) and zipped together with the Table/Dataset file.

<https://www.embopress.org/page/journal/17574684/authorguide#expandedview>

Thanks, we have worked to incorporate this structural change into the manuscript. Now, there are 6 main figures where all the source data are provided in the source data hierarchy. Out of the original supplemental figures, we have created 5 EV figures and the remaining 4 figures have become supplemental figures in the PDF appendix. The source data for the EV Figures and Supplemental Figures are also provided in the source data hierarchy.

12) Author contributions: CRediT has replaced the traditional author contributions section because it offers a systematic machine-readable author contributions format that allows for more effective research assessment. Please remove the Authors Contributions from the manuscript and use the free text boxes beneath each contributing author's name in our system to add specific details on the author's contribution. More information is available in our guide to authors.

Thank you. We have removed the CRediT section from the manuscript and will be sure to indicate these contributions on the checkbox when we re-submit.

13) Disclosure statement and competing interests: We updated our journal's competing interests policy in January 2022 and request authors to consider both actual and perceived competing interests. Please review the policy <https://www.embopress.org/competing-interests> and update your competing interests if necessary.

Thanks, no change was necessary.

14) Every published paper now includes a 'Synopsis' to further enhance discoverability. Synopses are displayed on the journal webpage and are freely accessible to all readers. They include a short stand first (maximum of 300 characters, including space) as well as 2-5 one-sentences bullet points that summarizes the paper. Please write the bullet points to summarize the key NEW findings. They should be designed to be complementary to the abstract - i.e. not repeat the same text. We encourage inclusion of key acronyms and quantitative information (maximum of 30 words / bullet point). Please use the passive voice. Please attach these in a separate file or send them by email, we will incorporate them accordingly.

Please note that these would be the final versions and changes during proofing are usually not allowed.

Thanks. We are submitting a separate file called Synopsis, with these elements included.

15) As part of the EMBO Publications transparent editorial process initiative (see our policy

here: https://www.embopress.org/transparent-process#Review_Process), Molecular Systems Biology will publish online a Peer Review File (PRF) to accompany accepted manuscripts. In the event of acceptance, this file will be published in conjunction with your paper and will include the anonymous referee reports, your point-by-point response and all pertinent correspondence relating to the manuscript. Let us know whether you agree with the publication of the PRF and as here, if you want to remove or not any figures from it prior to publication.

Please note that the Author checklist will be published at the end of the PRF.

Thanks for the heads-up; that is fine.

Molecular Systems Biology has a "scooping protection" policy, whereby similar findings that are published by others during review or revision are not a criterion for rejection. Should you decide to submit a revised version, I do ask that you get in touch after three months if you have not completed it, to update us on the status.

Yours sincerely,

Poonam Bheda, PhD
Scientific Editor
Molecular Systems Biology

Reviewer #1:

In this study, the authors ask a deceptively simple question: does the set of E. coli proteins that rely on cytosolic chaperones during their primary biogenesis in vivo match the set that need those same chaperones to refold from a fully-denatured state in vitro? To answer it they combine (i) quantitative limited-proteolysis mass-spectrometry (LiP-MS) on whole-cell lysates from strains lacking Trigger Factor (Tig) or Hsp70/Hsp40 (DnaKJ) with (ii) a rigorous re-analysis of their own 2022 proteome-wide refolding dataset. Selected enzymes (phosphoglycerate kinase, catalase) are purified to biochemically validate the structural proteomics calls.

Key Conclusions:

1) DnaKJ is required for native conformations of ~28% of the soluble proteome at 30 {degree sign}C but only ~17% at 37 {degree sign}C, whereas Tig deletion affects {less than or equal to}6% of proteins at either temperature.

2) Proteins that depend on DnaKJ in vivo show little overlap (13-17%) with proteins that require DnaKJ to refold in vitro, indicating distinct client sets for co-translational vs post-translational chaperone action.

3) Canonical "chaperone-non-refolders" such as PGK fold perfectly in vivo without Tig or DnaKJ, underscoring the power of vectorial, ribosome-guided folding.

4) DnaKJ dependence at low temperature is enriched for multi-domain proteins and holoproteins bearing covalently bound cofactors (heme, iron-sulfur, PLP). The authors propose that ATP-driven unfolding by DnaKJ helps these proteins escape kinetic traps that become self-resolvable at 37 {degree sign}C.

5) LiP-MS perturbations correlate with function: catalase (KatG) shows active-site structural changes in Δ dnaKJ lysates and a 35-40% drop in specific activity.

Overall, I really enjoyed reading this paper. This group continues to be at the forefront of MS-based proteostasis research and I appreciate the huge effort that went into this work.
Andy Truman

We thank the reviewer for their positive assessment of our paper. Moreover, they have nicely recapitulated what we also think are the main findings.

Minor comments:

1) GroEL is essential but highly induced in Δ dnaKJ. Could some DnaKJ clients be rerouted to GroEL? The authors should discuss this possibility.

We thank the reviewer for pointing this out. It is generally widely appreciated that knocking out such central chaperones like DnaKJ (or even trigger factor) is only tolerable because cells respond to these perturbations by rebalancing their proteostasis network, i.e., upregulating other chaperones and proteases. We did document (as have others) the large upregulation of GroEL in Fig S2A-B (now Fig EV2A-B). The reviewer is right to point out that

the text failed to mention that rerouting can (and almost certainly does) happen, implying that the structural changes we see are the ones that persist *even after* such rerouting occurs. We have added the following sentence [what's underlined is the addition. The previous paragraph is provided for context].

We find that deleting DnaKJ at 30°C results in a concomitant upregulation of other chaperones (GroEL, IbpB, IbpA, SurA) and proteases (Lon, ClpP, HslV), with many other (277) *E. coli* proteins' abundances going significantly down (>2-fold, Benjami-Hochberg adjusted p-value < 0.05; Fig. EV2 A). On the other hand, fewer chaperones and proteases have their levels dramatically altered when trigger factor is knocked out, and only 59 proteins' abundances significantly decrease (Fig. EV2 A-B). Because chaperone deletion causes the proteostasis network to rebalance (e.g., >4-fold higher levels of GroEL in $\Delta dnaKJ$), we note that the structural perturbations we observe here are those that persist even after other chaperones and proteases are upregulated, implying their native conformations require DnaKJ/trigger factor in a manner that is not readily complemented by other proteostasis factors.

The following sentence, in the original draft, also speaks to the idea that the reviewer is getting at.

Hence, even though trigger factor is an important chaperone that engages many *E. coli* nascent chains (Oh *et al*, 2011; Haldar *et al*, 2017; Martinez-Hackert & Hendrickson, 2009), these LiP-MS data suggest that its absence can be more easily compensated for by other members of the proteostasis network.

We also reinforce this point by adding the following sentence in the Discussion:

While GroEL is another critical chaperone in *E. coli*, it is essential and cannot be knocked out, which would make studying the structural consequences of its removal in living cells more challenging. We observe upregulation of GroEL in $\Delta dnaKJ$ cells, raising the possibility that some DnaKJ-clients may be rerouted to the GroEL/ES system in the absence of DnaKJ. It is likely that this shift helps mitigate the extent of structural perturbations in $\Delta dnaKJ$.

2) Fig. 2E-F: please consider using colour-blind-safe palettes as the red/green is indistinguishable.

This was a good suggestion. Now instead of assigning green to tig knock-out, it is universally reassigned to blue. So the primary colour palette for the key trends in the paper are based on a black-red-blue triad.

3) The abstract exceeds 250 words-I would recommend trimming this down a bit.

Good catch! The submission system actually stopped us from submitting for being over the limit, but we did not go back to change it in the manuscript itself. Here is the revised abstract at 160 words

Many proteins require molecular chaperones to fold into their functional native forms. However, the roles of chaperones during primary biogenesis *in vivo* can differ from the functions they play during *in vitro* refolding experiments. Here, we use limited proteolysis mass spectrometry (LiP-MS) to probe structural changes incurred by the *E. coli* proteome when two key chaperones, trigger factor and DnaKJ, are deleted. While knocking out DnaKJ induces pervasive structural perturbations across the soluble *E. coli* proteome, trigger factor deletion only impacts a small number of proteins' structures. Overall, proteins which cannot spontaneously refold (or require chaperones to refold *in vitro*) are *not* more likely to be dependent on chaperones to fold *in vivo*. We find that chaperone-nonfolders (proteins that cannot refold even with chaperone assistance) do not generally require chaperones to fold *in vivo*, strengthening the view that chaperone-nonfolders are obligate co-translational folders. Hence, for some *E. coli* proteins, the vectorial nature of co-translational folding is the most important "chaperone."

4) Check for consistent italics for gene names (dnaK, tig) and Roman for proteins (DnaK).

Thanks, we believe this has all been corrected.

5) Methods: please change "Eppendor Eporator" to "Eppendorf Eporator"

Thanks, corrected.

Reviewer #2:

In this work, the different roles of the DnaK and Trigger Factor (Tig) chaperones at helping the folding of newly synthesized proteins is analyzed *in vivo*, and compared with their roles in *in vitro* refolding experiments.

Overall, the manuscript is well-written, and it conveys a message that, although in my opinion less definitive than what the authors claim, represent a clear step forward and a somewhat provocative result.

Before I lay out my criticisms, I want to be clear on the fact that I am not asking for new experiments at this stage. My own rule is that new experiments should be performed only if necessary controls are lacking. In this case, the authors will be able to address my comments by casting their conclusions in a possibly milder way. Of course, should they be willing to perform the consequent experiments, they are most welcome, but I do not deem that necessary.

Here come my major comments, in no particular order:

1) I overall agree that the difference between 30C and 37C might be due to the increased "rigidity" of misfolded proteins at lower temperatures. At the same time, the authors mention these as "non-stress" temperatures. This should be given better context. Indeed, 37 C is physiological in the gut, but is 30 C physiological outside of it? Is *E. coli* adapted (non-

stress) over that window, or does it need some time to adapt as its environment changes (e.g. is ingested in or is expelled from the gut)? Likewise, other animals might have lower or higher physiological temperatures (e.g. birds are physiological at 39C, which can normally be used as a stressor in cultured *E. coli*; reptiles have likely a lesser body temperature). In this respect, the authors could use a word of caution when comparing results at the two temperatures to convey a message about the physiological roles.

This is an interesting point that the reviewer raises. Firstly, as a point of clarification, we would like to point out that nothing in these experiments were designed to interrogate how *E. coli* adapts to changes – it is not as though our 30°C experiments involved first growing cells at 37°C and then suddenly changing the temperature to 30°C (to create a “cold shock” as it were). Rather, the 30°C cells were incubated at 30°C overnight to create a saturated culture, subcultured to 0.05 OD, and then grown at 30°C back to log-phase (identically to the 37°C culture, except for them all steps were at 37°C). This is described in the Methods section.

Secondly, we would assert that 30°C *is* a non-stress condition for *E. coli*. It is a condition where it grows robustly (albeit somewhat slower than 37°C), and oftentimes to a higher maximal OD than it does at 37°C. So the statement that our work interrogates the dependency on chaperones during unstressed conditions does seem to us as an accurate and fair description.

We would agree with the reviewer, however, that 30°C (whilst unstressed) is still *not* a physiological condition for *E. coli*. So we have taken care to be more restricted in our use of the word “physiological condition”. This sentence was changed accordingly.

Taken together, these findings demonstrate that DnaKJ plays a critical role in shaping the structures of the native *E. coli* proteome under unstressed conditions, whereas trigger factor ostensibly plays a more specialized role that is also more easily compensated for by other chaperones.

We now do not use the word “physiological” to refer to the temperatures used in this study.

2) The authors have compared the DeltaDnaK and the DeltaTig with the WT, and looked for overlaps (with respect to the >1 cuts). Yet, since their role is not necessarily additive, an analysis of their direct comparison would have been also pretty telling. Any reasons not to do so?

This is an interesting point, but yes, there are a few reasons why we chose to make comparisons of DeltaDnaK and DeltaTig to a common reference (namely WT). The first has to do with the software we use for analyzing LiP-MS data, FLiPPR. To do a multi-condition comparison, FLiPPR can accept as many “test” conditions as the user supplies but will accept

only a single common reference that all test conditions are compared to. Adoption of a common reference condition is our preferred way to analyze LiP-MS data and we have applied this practice uniformly across our lab's body of work when performing multi-condition comparisons. In theory, we could perform a separate job altogether where, say, DeltaTig is assigned the reference, DeltaDnaK the test, and WT is omitted. However, because we actually get the most identifications in WT (see Fig EV1A, E), if we omitted WT, then our coverage would be decreased in the overall analysis because of the inability to match-between-runs to the WT samples. So the alternative analysis the reviewer proposes would work but would admit lower coverage.

3) The comparison with whole-proteome refolding (To et al, 2022) is very interesting, but what definitive statement can be made from it is less clear to me. In those experiments, the full proteome (lysate) was first denatured and then denaturant diluted while adding either GroELS or KJE. While this experiment tells about the intrinsic role of these chaperones, and as such is interesting of course, it meddles the waters a bit. Chaperones do not function, for the most part, as independent agents. KJE might hand-over its products to GroELS, or might be helped, in some cases such as sturdy aggregates, to ClpB (Hsp100). HtpG (Hsp90) might be relevant for a subset of substrates, such as kinases, refolding, while it might act independently or downstream of KJE. What about small heat shock proteins? While in the *in vitro* (lysate) experiments all of them were most likely denatured and absent, so that only the added chaperones were relevant, in the *in vivo* Deltas they are present in background, and I am not completely sure their effects can be easily disentangled.

Sure, this is a fair question. To clarify, the experiments that we completed in PNAS 2022 were indeed based on diluting a 6 M GdmCl-denatured lysate 100-fold with a buffer that was either supplemented with GroEL+ATP or DnaKJE+ATP. Since the endogenous chaperones in the *E. coli* lysate would have been denatured along with all the other proteins, and since *E. coli* chaperones are generally nonrefoldable, and since even if they could refold they would be present at ~nM concentrations in the diluted lysates, it is unlikely that any chaperone other than the one(s) that was/were supplemented would be "active" in the experiment. Hence, by comparing the proteins that refold *without* chaperones versus *with* a chaperone (Fig 5A, Fig EV5A) we obtain a measure of whether a protein is "chaperone-dependent" during *in vitro* refolding. This experimental design really does isolate the effect of a single chaperone system, even though, as the reviewer points out, chaperones oftentimes do work in tandem in the cell.

If the reviewer finds it surprising that 216 *E. coli* proteins appear to refold with DnaKJE alone (and without ClpB, HtpG, and small HSPs), we would point out that: (1) HtpG is a very specialized chaperone in *E. coli* with few known clients (quite unlike Hsp90s in eukaryotes); (2) Small heat shock proteins are important primarily to prevent aggregation which we mitigate in these experiments simply by refolding at very low overall concentrations; and (3)

ClpB in *E. coli* is a disaggregase that functions primarily during severe heat stress, so would not be as relevant under conditions where aggregation is suppressed.

On the other hand, the primary dataset that is presented in this paper shows structural perturbations when a chaperone is knocked out and all other chaperones preserved *in vivo*. Indeed, under these genetic conditions, other chaperones are generally upregulated. So for a protein to be structurally perturbed in the present assays, they would have had to have been expressly dependent on the chaperone(s) that was/were knocked out, in a manner that could not be compensated for by other chaperones (a point we have reinforced in the paper, thanks to a comment raised by reviewer 1).

So in essence, these two datasets comprise a list of proteins that are chaperone-dependent during *in vitro* refolding and a list of proteins that are chaperone-dependent during *in vivo* biogenesis. The primary result of Fig 5B-C is that these two lists do not overlap very much.

Perhaps one reason why this point is subtle/confusing, is that our definition of chaperone-dependence *in vivo* is based on one LiP-MS experiment (which compares WT to KO), but our definition of chaperone-dependence *in vitro* is actually based on comparing *two* LiP-MS experiments (namely, native-to-refolded without chaperones *to* native-to-refolded with chaperones). This is exactly why in Fig 5, we spell this out with the contingency-table and the colour code. We agree with the reviewer it is a subtle point, but we stand by the assertion that our experiments, collectively, can compare chaperone dependence *in vitro* (from PNAS 2022) to *in vivo* (the present work).

4) What said at point 3 applies even more for "non-refoldable" substrates. They were deemed non-refoldable when the denatured lysate was supplemented with either KJE or with GroEL. What if they needed the full chaperone network to be refoldable, including possibly being captured by sHSPs along the way?

It is absolutely correct that our studies do not directly probe the effect of multi-chaperone dependency because in PNAS 2022 we only supplemented a single chaperone system into the reaction, and in the present work we only deleted at most a single chaperone system.

Let's perform a thought-experiment of what the outcome of the assays would have been for a hypothetical client that uses both DnaKJE and GroEL/ES (or for that matter, DnaKJE and *any other chaperone*). If it needed both chaperones *in vivo*, then it would have been structurally perturbed in $\Delta dnaKJ$. If it needed both chaperones during *in vitro* refolding, then it would have been in the "nonrefolding under both conditions" category in Fig 5A (which has 124 members). The analysis in Fig 5B-C currently does not focus on this category, but only on the ones which we *could* get to refold *in vitro* with DnaK.

To clarify this, we have added extra detail to a parenthetical note in the text:

When cross-correlating refolding experiments without chaperones (whose overall refolding rate was 58.2%, 603 out of 1029 proteins) to refolding experiments with DnaKJ and GrpE (whose overall refolding rate was 77.5%, 564 out of 728 proteins), we found that 701 could be confidently assessed in both, of which 216 were DnaK-dependent refolders (that is, they refolded with DnaKJ but not without; note this set would *not* include proteins that require DnaKJ along with other chaperones; Fig. 5A)

Now, let's go further. We do have 124 proteins which did not refold even with DnaKJ (see Fig 5A). What if we cross-correlated *those* against our set of structurally perturbed proteins in $\Delta dnaKJ$ (DnaKJ-dependent *in vivo*)? There are two outcomes. Either these 124 proteins were, for the most part, proteins that required DnaKJ+another chaperone; *or*, they are proteins that required co-translational folding. If it were the former, then we should find that these 124 proteins overlap strongly with the structurally-altered-in- $\Delta dnaKJ$ set. But, alas, they still don't (see below)!

So this provides evidence that DnaKJ-nonrefolders more likely require co-translational folding rather than additional chaperones to fold.

We performed this extra analysis for the reviewer, but opted to withhold this from the revised manuscript because the analysis is very similar (almost redundant) to what is already in the paper as Fig 5E-I. This is because the set of 124 DnaKJ-nonrefolders is very similar to the set of 95 "chaperone-nonrefolders" (which we define in Fig 5E).

5) I take some issue with the physiological relevance of refolding experiments, if the goal is to infer the role of chaperones. Let me be clear: To et al. 2022 is an excellent paper. But in the context of the present one, I am not so sure that it is the good way to argue about post-translational refolding (but to be fair: this is a criticism for 95% of the experiments in the literature). This is my beef with it:

I doubt that translated and folded proteins do suddenly unfold, unchecked, and then chaperones have to cope with them. Most likely, chaperones act immediately at the first signs of unfolding/misfolding, and maintain proteins native possibly even in mildly non-native conditions, in particular in the non-stressful region that the authors have investigated

here.

I would like to authors to at least comment about this.

We heartedly agree with the reviewer on the points raised in the paragraph above.

Firstly – yes – we agree that *in vitro* refolding experiments are not really physiologically relevant. Cells never see 6 M GdmCl. Globular proteins in cells (likely) never populate fully denatured states. We make this point in the introduction:

However, these experiments do not provide information about the role of *E. coli* chaperones during primary protein biogenesis, which occurs co-translationally... In addition, cells very rarely see fully denatured protein conformations of the kind that would be populated by high concentrations of denaturant, even during acute heat shock, because sub-lethal heat shock temperatures are still considerably lower than typical T_m 's of many mesophilic protein...

The point is reiterated in the introduction to the "**Comparisons of Chaperone Dependence *in vivo* and *in vitro***" section:

Although unrelated to their main functions *in vivo*, chaperones also assist the refolding of proteins that have been fully unfolded by denaturant *in vitro*...

So, the lack of "physiological relevance" to the classic denaturant-induced-refolding experiment is indeed, very likely why we obtained the result in Fig 5B-C (of low overlap between chaperone dependency *in vitro* and *in vivo*) that we did.

The notion that proteins do not suddenly become fully unfolded (as they would do in 6 M GdmCl) – either during synthesis or during stress – is the point we are trying to make in the "counterfactual" scenario depicted in Fig 5D. We have tried to clarify our discussion of Fig 5D, along the lines the reviewer is discussing, by adding the following text:

In other words, the data suggest that the proteins which require DnaKJ *in vivo* during primary biogenesis are generally unrelated to the proteins which require DnaKJ *in vitro* during total refolding. The only obvious reason why this could be is that *in vivo* DnaKJ primarily acts co-translationally and interacts with a distinct set of intermediates, which are partially unfolded/misfolded rather than fully unfolded.

Based on the tone of the reviewer, we do wonder if perhaps we were not sufficiently clear that our depiction in Fig 5D was indeed meant to be a counterfactual scenario! We have made sure this is pointed out in the revised Fig 5D.

6) Last but not least: How can the authors clearly disentangle the misfolding of newly synthesized proteins from the one of proteins that are already present? What I mean is that should KJ also act as suggested in point 5, there could be some native state maintenance in the WT that is lacking in the DeltaKJ, so that there might be some mixture of co-

translational and post-translational misfolding. Experiments to test this could be difficult but not impossible (like starting feeding isotope-modified cultures at some point, to distinguish newly synthesized proteins from the pre-existing ones). I am not asking for them, but maybe some discussions would be useful.

Outside of stress conditions, one can imagine each protein existing in one of three stages: (1) as a nascent chain during co-translational folding (these can be probed with specialized selective ribosome profiling, SeRP, experiments); (2) immediately post-translational, i.e., newly-synthesized (these can be probed through pulse isotope labeling); and (3) post-translational (what we typically think of that gets probed during standard proteomics).

Elegant work from Mayor's lab performed an assay similar to what the reviewer described in their Cell Reports **2022** paper (DOI: [10.1016/j.celrep.2022.111096](https://doi.org/10.1016/j.celrep.2022.111096)). Our assay of course does not explicitly remove nascent and newly-synthesized protein. However, it is very likely that our data do report on the third (post-translational) stage to the exclusion of the other two categories. Nascent chains are pretty scarce. There are 40,000 ribosomes/E. coli cell, so at most 40,000 nascent chains. However there are $\sim 2 \times 10^6$ proteins/E. coli cell, so nascent chains make up $\sim 2\%$ of the proteome. The time that we might call a protein "newly-synthesized" is somewhat arbitrary, but let's say that it is equal to the time it took to synthesize the protein itself, after translation. For E. coli, this is on average 15 s. In E. coli proteins are rarely actively degraded so their "total lifetime" can be estimated at the doubling time of a cell, or 20 min. So the newly-synthesized proteome is $\sim 1\%$ of the mass of the proteome, leaving 97% of the mass to be post-translational.

Now, we do not actively remove nascent chains or newly-synthesized proteins. But they would only contribute 3% to the total material we collect. To label a peptide structurally altered under one of our conditions, a peptide needs to have >2 -fold change in its signal between WT and KO by our cut-offs. Hence, it is unlikely a conformational change from a subpopulation that comprises 3% of the ensemble would result in an observable change. In other words, our studies are insensitive to structural changes induced by Δ dnaKJ or Δ tig that occur on nascent proteins or newly-synthesized proteins, only to structural changes that endure in the bulk proteome.

The reviewer may find it interesting though that the Bukau lab did recently publish a systematic SeRP study to look at where chaperones bind nascent proteins co-translationally, and encouragingly we do find that proteins which are structurally perturbed in Δ dnaKJ possess higher "binding scores" to DnaKJ co-translationally (now shown in Fig EV5D-E). So this observation does support the view that under non-stress conditions, DnaKJ is important co-translationally and its absence during protein synthesis results in structural changes that persist following ejection from the ribosome. This point is now made in the new version of the manuscript (see also response to Reviewer 3, comment 1 to see this new paragraph).

Minor:

here and there some passages could be smoother. Please re-read carefully with the eyes of the audience, not of the authors.

We have endeavoured to make the manuscript broadly understandable and have made a few light changes for style and clarity throughout in the revision. We do concede that the points made in Figure 5 are pretty subtle/complex and may require readers have familiarity with our previous work to fully grasp it. On the other hand, we also think there are a number of 'simpler' points the paper makes on its own that do not rely on such familiarity, so we view the paper as having "something for everyone": Figs 2-4 for a broad audience and Fig 5 for chaperone aficionados.

Fig.6E at page 24 should be 5E, if I am not mistaken.

Thanks for catching, we have fixed accordingly.

Reviewer #3:

The protein folding problem is one of the fundamental questions in biology. Conventional folding studies, originating from Anfinsen's classical experiments, do not always reflect folding in vivo, since nascent polypeptides are synthesized vectorially from the N-terminus. This manuscript clearly demonstrates the difference between in vivo and in vitro folding, using a large-scale analysis with limited proteolysis mass spectrometry (LiP-MS). The study presented here is exactly what I hoped to see following the author's series of works using LiP-MS analyses. The results are highly interesting, and I have a few comments to clarify unclear points or to further strengthen the current conclusions.

1. If the group of structurally perturbed proteins in vivo, shown in this manuscript, is indeed cotranslationally assisted in folding by chaperones, especially DnaK, then these chaperones would bind to the ribosome during translation via the nascent chain. In that case, one could expect a correlation between the protein groups in this study and the proteome-wide data obtained through selective ribosome profiling, which would allow for a comprehensive analysis of the binding of TF and DnaK to nascent chains (Galmozzi et al., 2025, cited in this manuscript). I encourage the authors to analyze this to strengthen the conclusions of this study.

Thanks for this comment. We have looked at the "binding scores" that Galmozzi et al. tabulated for nascent chains to DnaK, and indeed, encouragingly we do find that proteins which are structurally perturbed in Δ dnaKJ possess higher "binding scores" to DnaKJ cotranslationally (now shown in Fig EV5D-E). Here is the text of the new paragraph describing the results.

By focusing on the actual structural outcome of protein biogenesis, rather than on chaperone binding (which can be measured by co-precipitation mass spectrometry (Kerner *et al*, 2005; Calloni *et al*, 2012)) and by selective ribosome profiling (SeRP, Oh *et al*, 2011; Shiber *et al*, 2018; Stein *et al*, 2019; Galmozzi *et al*, 2025)), we can see that chaperones sometimes are involved in folding processes without being required to achieve natively-folded structures. Although our experiments do not directly report on the structure of nascent chains or newly-synthesized proteins, we can compare our results to a recent SeRP study that assessed which nascent proteins engage *E. coli* chaperones co-translationally (Data ref: Galmozzi *et al*. 2025). We find that proteins which become structurally perturbed in $\Delta dnaKJ$ indeed interact more with DnaKJ as nascent chains ($p < 3 \times 10^{-4}$ by Kolmogorov–Smirnov test; Fig. EV5D-E, see Source Data for Fig. EV5). This observation supports the view that under non-stress conditions, DnaKJ supports co-translational folding, and its absence during protein synthesis results in structural changes that persist following ejection from the ribosome. By contrast, we did not find that proteins which become structurally perturbed in Δtig possessed greater interaction with trigger factor as nascent chains (Fig. EV5F-G). This result could either be simply due to less statistical power from having fewer examples, or alternatively be construed as showing that trigger factor binds to many nascent proteins but is not essential for acquisition of native structure.

2. The finding that structurally perturbed proteins in the $\Delta dnaKJ$ strain are more prominent at lower temperature (30C) is indeed surprising.

We think so too. Hopefully we have successfully managed to convey this observation and provide a credible potential explanation in the form of Fig 6 and the discussion points around it.

2-a. Is there any difference in the perturbed proteins between 30C and 37C in wild-type *E. coli*?

This is a good question, which we can address by setting up new LFOs comparing the WT strain at 37°C to the same (WT) strain at 30°C. The results from this analysis are now shown in Figure S1 (in the Appendix PDF) and are described in the main text, at the end of the first Results section:

We also examined whether temperature alone alters protein structure in the wild-type background. Comparing WT proteomes at 30 °C and 37 °C by LiP-MS revealed 66 proteins out of 1434 (4.6%) with altered PK susceptibility, each possessing two or more significant cut sites (Appendix Fig. S1A-B, Source Data Figure S1). The most highly perturbed was RaiA, a ribosome hibernation factor, with the largest number of altered cut sites (19), consistent with its role in modulating ribosome function under environmental changes. Other affected categories included RNA maintenance factors (DeaD [also called cold-shock DEAD-box protein] and Rne), three chaperones (DnaK, ClpB, HtpG), and seven proteins involved in polyatomic anion redox chemistry.

2-b. Proteins that aggregate or are degraded at 37C in the *dnaKJ*-KO strain may be more abundant than under 30C conditions. Are there any effects of these?

This is an interesting suggestion. We ran a new LFQ in which we calculated protein abundance ratios (so, using the Trypsin-only data, *not* LiP) in 37°C/30°C both in $\Delta dnaKJ$, and we also would have expected what the reviewer said: That given that 37°C is a more pro-aggregation/pro-degradation condition, protein abundances would overall be lower at 37°C in $\Delta dnaKJ$. The data did not show this; instead, there were more proteins with higher levels at 37°C. Moreover, the specific proteins at the “extremes” (much more abundant in 37°C or 30°C) did not “jump out” to us as having discernible trends or commonalities. As a consequence, we have opted to not include this additional analysis in the paper.

Minor points:

3. p. 24: Fig. 6E should be Fig. 5E.

Good catch. Fixed.

4. p. 25: CsdA should be CspA.

Good catch. Actually, we changed it to DeaD, because now this point is quite relevant to the discussion to the 30°C vs. 37°C comparison we did at the reviewer’s request. EcoCyc seems to suggest DeaD is the canonical name, and CsdA is a synonym.

15th Sep 2025

Manuscript Number: MSB-2025-13201R

Title: Chaperone Dependency during Biogenesis Does Not Correlate with Chaperone Dependency during Refolding

Dear Prof Fried,

Thank you for the submission of your revised manuscript to Molecular Systems Biology. We have now received the enclosed reports from the referees that were asked to re-assess it. As you will see the reviewers are now globally supportive and I am pleased to inform you that we will be able to accept your manuscript pending the following final amendments:

- 1) In the main manuscript file, please reduce keywords to max. 5.
- 2) Please rename the "Data and materials availability" section to "Data Availability" and format according to the example below: "The datasets and computer code produced in this study are available in the following databases:
 - Chip-Seq data: Gene Expression Omnibus GSE46748 (<https://www.ncbi.nlm.nih.gov/geo/query/acc.cgi?acc=GSE46748>)
 - Modeling computer scripts: GitHub (<https://github.com/SysBioChalmers/GECKO/releases/tag/v1.0>)
 - [data type]: [full name of the resource] [accession number/identifier] ([doi or URL or identifiers.org/DATABASE:ACCESSION])"
- 3) Please ensure that the proteomics data in PRIDE is now publicly released (and remove the "Reviewer access details" section in the manuscript).
- 4) Please rename "Declaration of competing interest" to "Disclosure and competing interests statement". We updated our journal's competing interests policy in January 2022 and request authors to consider both actual and perceived competing interests. Please review the policy <https://www.embopress.org/competing-interests> and update your competing interests if necessary.
- 5) Author contributions: Please remove it from the manuscript and specify author contributions in our submission system. CRediT has replaced the traditional author contributions section because it offers a systematic machine-readable author contributions format that allows for more effective research assessment. You are encouraged to use the free text boxes beneath each contributing author's name to add specific details on the author's contribution. More information is available in our guide to authors: <https://www.embopress.org/page/journal/17574684/authorguide#authorshipguidelines>
- 6) Please rename the "References Cited" section to "References".
- 7) In the Methods, please take care of the following:
 - The Materials and Methods section should be renamed to "Methods".
 - Please ensure that a statement on whether or not blinding was done is included in the Methods even if no blinding was done. Please also be sure to update the Author Checklist with this information and where it can be found in the manuscript.
- 8) Please place individual sections of the manuscript in the following order: Title page - Abstract & Keywords - Introduction - Results - Discussion - Methods - Data Availability - Acknowledgements - Disclosure and Competing Interests Statement - References - Figure Legends - Expanded View Figure Legends.
- 9) For the figures and figure legends, please take care of the following:
 - Please make sure to update the callouts of all figures in the main manuscript text so that they are called out sequentially. Also callouts for Fig. 6A-B and Appendix Table S1 are missing.
 - Please note that the legends for figure EV5 is not provided in the sequential manner. This needs to be rectified.
 - Please note that information related to n is missing in the legends of figures 4B, EV4 D, EV5 D-G.
- 10) In the Appendix file, the nomenclature should be Appendix Figure Sx and Appendix Table Sx throughout the manuscript and Appendix PDF.
- 11) Please remove the synopsis image from the synopsis text file and upload it separately as a high-resolution jpeg file, ensuring that the dimensions are 550 pixels wide x (300-600) pixels high.
- 12) Source Data: Please ensure that a completed Source Data checklist is uploaded as a Related Manuscript File. Source Data should be organized as a single source data file (zipped) per figure for main figures (all EV and/or Appendix figure Source Data can be included in a single folder), with the panels clearly visible in the folder structure instead of a single excel file for all Source Data. e.g. all the Source data files for figure 1 need to be saved in a single folder and this needs to be zipped and then uploaded as "SD figure 1.zip" file.
- 13) As part of the EMBO Publications transparent editorial process initiative (see our policy here: https://www.embopress.org/transparent-process#Review_Process), Molecular Systems Biology will publish online a Peer Review File (PRF) to accompany accepted manuscripts. This file will be published in conjunction with your paper and will include the anonymous referee reports, your point-by-point response and all pertinent correspondence relating to the manuscript. Let us know whether you agree with the publication of the PRF and as here, if you want to remove or not any figures from it prior to publication. Please note that the Authors checklist will be published at the end of the PRF.
- 14) After your paper is published, we may promote it on social media. If you have any handles or hashtags for Bluesky you would like included, please let us know.
- 15) Please provide a point-by-point letter INCLUDING my comments and your detailed responses (as Word file).

I look forward to reading a new revised version of your manuscript as soon as possible.

Yours sincerely,

Poonam Bheda, PhD
Scientific Editor
Molecular Systems Biology

Reviewer #1:

The authors have put substantial effort into addressing my (and the other reviewers) comments and in doing do have produced a highly improved manuscript. I congratulate Dr. Fried and team on an innovative and exciting paper that makes major strides in understanding proteostasis. I highly recommend this work for publication. Andy Truman

Reviewer #2:

While I do not share the same confidence of the authors about the full conclusiveness of their findings, I find their replies to my concerns stimulating, mostly correct or anyway strong enough to be a robust foundation, rich in downstream questions, upon which research by the same group, or others, could (and ideally should) continue. I am thus satisfied by the way they addressed the first round of review.

Reviewer #3:

The authors adequately addressed all the concerns I raised. I am fully satisfied with the revision.

Dear Prof Fried,

Thank you for the submission of your revised manuscript to Molecular Systems Biology. We have now received the enclosed reports from the referees that were asked to re-assess it. As you will see the reviewers are now globally supportive and I am pleased to inform you that we will be able to accept your manuscript pending the following final amendments:

1) In the main manuscript file, please reduce keywords to max. 5.

Done.

2) Please rename the "Data and materials availability" section to "Data Availability" and format according to the example below:

"The datasets and computer code produced in this study are available in the following databases:

- Chip-Seq data: Gene Expression Omnibus GSE46748

(<https://www.ncbi.nlm.nih.gov/geo/query/acc.cgi?acc=GSE46748>)

- Modeling computer scripts: GitHub

(<https://github.com/SysBioChalmers/GECKO/releases/tag/v1.0>)

- [data type]: [full name of the resource] [accession number/identifier] ([doi or URL or identifiers.org/DATABASE:ACCESSION])"

Done.

3) Please ensure that the proteomics data in PRIDE is now publicly released (and remove the "Reviewer access details" section in the manuscript).

PRIDE requires a DOI or PMID in order to make the data public. We can do this as soon as you provide the paper's DOI.

4) Please rename "Declaration of competing interest" to "Disclosure and competing interests statement". We updated our journal's competing interests policy in January 2022 and request authors to consider both actual and perceived competing interests. Please review the policy <https://www.embopress.org/competing-interests> and update your competing interests if necessary.

We changed the heading but did not change the text below it, as we neither have actual nor perceived competing interests.

5) Author contributions: Please remove it from the manuscript and specify author contributions in our submission system. CRediT has replaced the traditional author contributions section because it offers a systematic machine-readable author contributions format that allows for more effective research assessment. You are encouraged to use the free text boxes beneath each contributing author's name to add specific details on the author's contribution. More information is available in our guide to authors:

<https://www.embopress.org/page/journal/17574684/authorguide#authorshipguidelines>

The last version already had the author contributions section removed, and we have adopted the CRediT system. I will look for where we can specify it on the submission system. The last version had an acknowledgement section which acknowledges non-authors' contributions, which we would like to keep.

6) Please rename the "References Cited" section to "References".

Done

7) In the Methods, please take care of the following:

- The Materials and Methods section should be renamed to "Methods".

Done

- Please ensure that a statement on whether or not blinding was done is included in the Methods even if no blinding was done. Please also be sure to update the Author Checklist with this information and where it can be found in the manuscript.

We now preface the Methods section with a statement "Experiments in this study were conducted without blinding."

8) Please place individual sections of the manuscript in the following order: Title page - Abstract & Keywords - Introduction - Results - Discussion - Methods - Data Availability - Acknowledgements - Disclosure and Competing Interests Statement - References - Figure Legends - Expanded View Figure Legends.

This is now observed although we also have a section in between Disclosure and References called "Supplementary material" which we think we found in other MSB papers that had an appendix PDF. You have our permission to delete this if you do not want it to appear in the final paper.

9) For the figures and figure legends, please take care of the following:

- Please make sure to update the callouts of all figures in the main manuscript text so that they are called out sequentially.

Thanks for asking, they are sequential.

Also callouts for Fig. 6A-B and Appendix Table S1 are missing.

Thanks. The one call-out to Fig. 6 has been replaced with "Fig. 6A-B"

In the methods section, we include an extra sentence "Oligonucleotides used in these manipulations are listed in Appendix Table S1."

- Please note that the legends for figure EV5 is not provided in the sequential manner. This needs to be rectified.

Thanks. Fixed.

- Please note that information related to n is missing in the legends of figures 4B, EV4 D, EV5 D-G.

Thanks. Fixed in each case.

10) In the Appendix file, the nomenclature should be Appendix Figure Sx and Appendix Table Sx throughout the manuscript and Appendix PDF.

Thanks. Fixed.

11) Please remove the synopsis image from the synopsis text file and upload it separately as a high-resolution jpeg file, ensuring that the dimensions are 550 pixels wide x (300-600) pixels high.

Thanks. Fixed.

12) Source Data: Please ensure that a completed Source Data checklist is uploaded as a Related Manuscript File.

When we last tried to do this, we found that the file system would not properly upload the source data checklist and hence provided it to you over email instead. We will try to upload it again and will also include it as an attachment to this email if, again, the upload of the file to the submission portal does not work.

Source Data should be organized as a single source data file (zipped) per figure for main figures (all EV and/or Appendix figure Source Data can be included in a single folder), with the panels clearly visible in the folder structure instead of a single excel file for all Source Data. e.g. all the Source data files for figure 1 need to be saved in a single folder and this needs to be zipped and then uploaded as "SD figure 1.zip" file.

We are a little confused by this comment, because we did organize the source data exactly as you described in the previous submission. If the naming of the zip file (we did not use the naming convention you describe) was the issue, then we have rectified by renaming the zip files according to the convention you recommend.

13) As part of the EMBO Publications transparent editorial process initiative (see our policy here: https://www.embopress.org/transparent-process#Review_Process), Molecular Systems Biology will publish online a Peer Review File (PRF) to accompany accepted manuscripts. This file will be published in conjunction with your paper and will include the anonymous referee reports, your point-by-point response and all pertinent correspondence relating to the manuscript. Let us know whether you agree with the publication of the PRF and as here, if you want to remove or not any figures from it prior to publication. Please note that the Authors checklist will be published at the end of the PRF.

That is all fine, you can publish the PRF.

14) After your paper is published, we may promote it on social media. If you have any handles or hashtags for Bluesky you would like included, please let us know.

Thanks! The relevant handles are @fried_lab (on X) and @friedlab.bsky.social (on BlueSky)

15) Please provide a point-by-point letter INCLUDING my comments and your detailed responses (as Word file).

See above. And we will also include a word file copy in the final submission to the portal.

26th Sep 2025

Manuscript number: MSB-2025-13201RR

Title: Chaperone Dependency during Biogenesis Does Not Correlate with Chaperone Dependency during Refolding

Dear Prof Fried,

Congratulations on an excellent manuscript, I am pleased to inform you that your manuscript has been accepted for publication in Molecular Systems Biology. Thank you for your comprehensive response to referee concerns. We have thus commissioned a News & Views to highlight your work, which should be published in MSB at the same time as your paper. It has been a pleasure to work with you to get your paper to the acceptance stage.

Yours sincerely,

Poonam Bheda, PhD
Scientific Editor
Molecular Systems Biology
